# The oxygen sensor prolyl hydroxylase domain 2 regulates the in vivo suppressive capacity of regulatory T cells

**Yousra Ajouaou[1,2], Abdulkader Azouz[1,3], Anaëlle Taquin[1,2], Sebastien Denanglaire[1,2], Hind Hussein[1,2], Mohammad Krayem[4,5], Fabienne Andris[1,2], Muriel Moser[1,2], Stanislas Goriely[1,2,3], Oberdan Leo[1,2]\***

[1]U-CRI (ULB Center for Research in Immunology), Université Libre de Bruxelles (ULB), Charleroi, Belgium; [2]Immunobiology Laboratory, Université Libre de Bruxelles (ULB), Charleroi, Belgium; [3]Institute for Medical Immunology, Université Libre de Bruxelles (ULB), Charleroi, Belgium; [4]Department of Radiation Oncology, Institut Jules Bordet, Université Libre de Bruxelles, Brussels, Belgium; [5]Laboratory of Clinical and Experimental Oncology (LOCE), Institut Jules Bordet, Université Libre de Bruxelles, Brussels, Belgium

**Abstract** The oxygen sensor prolyl hydroxylase domain 2 (PHD2) plays an important role in cell hypoxia adaptation by regulating the stability of HIF proteins (HIF1α and HIF2α) in numerous cell types, including T lymphocytes. The role of oxygen sensor on immune cells, particularly on regulatory T cell (Treg) function, has not been fully elucidated. The purpose of our study was to evaluate the role of PHD2 in the regulation of Treg phenotype and function. We demonstrate herein that selective ablation of PHD2 expression in Treg (PHD2$^{\Delta Treg}$ mice) leads to a spontaneous systemic inflammatory syndrome, as evidenced by weight loss, development of a rectal prolapse, splenomegaly, shortening of the colon, and elevated expression of IFN-γ in the mesenteric lymph nodes, intestine, and spleen. PHD2 deficiency in Tregs led to an increased number of activated CD4 conventional T cells expressing a Th1-like effector phenotype. Concomitantly, the expression of innate-type cytokines such as *Il1b*, *Il12a*, *Il12b*, and *Tnfa* was found to be elevated in peripheral (gut) tissues and spleen. PHD2$^{\Delta Treg}$ mice also displayed an enhanced sensitivity to dextran sodium sulfate-induced colitis and toxoplasmosis, suggesting that PHD2-deficient Tregs did not efficiently control inflammatory response in vivo, particularly those characterized by IFN-γ production. Further analysis revealed that Treg dysregulation was largely prevented in PHD2-HIF2α (PHD2-HIF2α$^{\Delta Treg}$ mice), but not in PHD2-HIF1α (PHD2-HIF1α$^{\Delta Treg}$ mice) double KOs, suggesting an important and possibly selective role of the PHD2-HIF2α axis in the control of Treg function. Finally, the transcriptomic analysis of PHD2-deficient Tregs identified the STAT1 pathway as a target of the PHD2-HIF2α axis in regulatory T cell phenotype and in vivo function.

**\*For correspondence:**
oberdan.leo@ulb.be

**Competing interest:** The authors declare that no competing interests exist.

## Editor's evaluation

The possibility that hypoxia signaling pathways exert control over immunity by actions on regulatory T lymphocytes is of great interest given that immune inflammatory pathology is often associated with hypoxia. This article supports the cell-intrinsic role of a specific isoform of hypoxia-inducible factor 2 (HIF2) in reducing the function of these cells when HIF is activated by inactivation of its regulatory hydroxylase. Aside from the biology, this is important because both HIF activators and specific HIF-2 inhibitors are in use clinically.

## Introduction

CD4[+] regulatory T cells (Tregs), accounting for approximately 5–10% of total circulating CD4[+] T cells, represent a critical subset of T lymphocytes involved in immune homeostasis. Through a broad set of effector mechanisms, these cells contribute to immune tolerance to self-constituents and to mucosal antigens derived from the commensal microflora and food (*Lu et al., 2017*; *Xing and Hogquist, 2012*). Tregs also participate in the resolution of inflammatory responses (*Dominguez-Villar and Hafler, 2018*) and play an important role in maternal immunotolerance against the semi-allogeneic fetus (*Samstein et al., 2012*). While critical to maintain tissue integrity, excessive activation of Tregs impedes adequate immune responses to tumors and pathogens, suggesting a tight control of their suppressive activity and tissue localization (*Togashi et al., 2019*; *Aandahl et al., 2004*; *van der Burg et al., 2007*). Although uniformly characterized by the expression of the lineage-specific, Foxp3 transcription factor (*Lu et al., 2017*), regulatory T cells display a wide range of phenotypic and functional properties that allow them to migrate to specific sites and suppress a variety of immune reactions, including inflammatory (*Caridade et al., 2013*) and humoral responses (*Clement et al., 2019*). Although diverse, the mechanisms whereby Tregs antagonize the activity of immune effectors are largely paracrine in nature (*Vignali et al., 2008*). Short-range suppressive mechanisms include competition for nutrients and/or growth factors (mostly cytokines), secretion of immunosuppressive factors, and direct, contact-mediated, inactivation of antigen-presenting cells (*Wardell et al., 2021*). These findings suggest that Tregs have to adapt to multiple lymphoid and nonlymphoid environments and suppress immune responses in a context and tissue-dependent fashion (*Shevyrev and Tereshchenko, 2019*; *Panduro et al., 2016*).

Oxygen represents an essential component of cellular bioenergetics and biochemistry. Because oxygen tension varies according to tissues and pathophysiological states (*Ast and Mootha, 2019*), cells need to adapt to fluctuations in oxygen availability in order to maintain an adequate functional and metabolic status. Of note, low-oxygen availability (hypoxia) plays a critical role in the pathophysiology of many immune disorders (*McKeown, 2014*; *Bartels et al., 2013*). Inflammation, in particular, is thought to reduce oxygen availability to tissues by affecting microvascular form and function, and through the recruitment of highly oxygen-consuming inflammatory cells producing NADPH oxidase-derived reactive oxygen species (*Nanduri et al., 2015*).

Immune cells patrolling through lymphoid and nonlymphoid tissues need therefore to readily adapt to varying oxygen concentration levels in order to exert their function (*Mempel and Marangoni, 2019*), suggesting an important role for oxygen sensors in immune regulation. Several hypoxia-sensitive pathways are known to enable single-cell survival in low-oxygen settings. In particular, reduced oxygen levels are directly sensed by a family of oxygen-dependent prolyl hydroxylases (PHD encoded by *Egln gene*, comprising three members) (*Hirota, 2020*), although other mechanisms mediated by oxygen-sensitive histone lysine demethylases (KDM) and the cysteamine dioxygenase/N-degron pathway have been recently uncovered (*Baik and Jain, 2020*). Hypoxia-inducible factors (HIFs), a set of evolutionary conserved transcriptional regulators, represent the best-described substrates of PHDs. These factors are heterodimers composed of a HIFα subunit whose stability is directly controlled by oxygen availability and a constitutively expressed HIF1β subunit (also known as ARNT) (*Wang and Semenza, 1995*). Following the initial characterization of the first member of HIFα family (HIF1α), two additional members, HIF2α and HIF3α, have been identified and shown to be similarly regulated by $O_2$ availability and bind to HIF1β (*Webb et al., 2009*). In normoxia, PHDs, whose affinity for oxygen is low and comparable to atmospheric concentrations, catalyze the prolyl-hydroxylation of HIFα subunits (*Semenza, 2001*). This post-translational modification allows recognition and ubiquitination of HIFα subunits by the E3 ubiquitin ligase Von Hippel–Lindau protein (pVHL) and subsequent degradation by the proteasome (*Maxwell et al., 1999*). In hypoxia, nonhydroxylated alpha subunits escape degradation and translocate to the nucleus where they bind to constitutively expressed and stable beta subunits to constitute an active heterodimer able to regulate gene expression. This process promotes transcriptional regulation of numerous genes, ultimately leading to increased oxygen supply (such as angiogenesis) and promotion of anaerobic metabolism (*Downes et al., 2018*).

Multiple levels of complexity of this major regulatory axis have been recently uncovered. As previously suggested, the presence of several members of the PHD and HIF families suggests specialized functions of PHD-HIF pairs during ontogeny and in selected tissues (*Watts and Walmsley, 2019*). In particular, while HIF1α appears as ubiquitously expressed in all metazoans, HIF2α represents a late

acquisition of vertebrates, displaying a more restricted tissue expression pattern (*Talks et al., 2000*). Although these factors bind to similar sequence motifs (hypoxia response elements [HREs]) and regulate the expression of a shared set of genes, both HIF1α and HIF2α-specific gene targets have been identified in multiple tissues (*Downes et al., 2018*; *Hu et al., 2007*; *Bono and Hirota, 2020*). Further complexity in this pathway stems from the possible occurrence of additional, non-HIF-related, PHD substrates (*Meneses and Wielockx, 2016*; *Mikhaylova et al., 2008*; *Chan et al., 2009*; *Romero-Ruiz et al., 2012*; *Huo et al., 2012*; *Xie et al., 2015*; *Lee et al., 2015b*; *Guo et al., 2016*), a set of findings that however has not been confirmed in more rigorous in vitro settings using well-defined synthetic substrates (*Cockman et al., 2019*).

Hypoxia plays a dual role in inflammation and in the regulation of immune responses. In most settings, hypoxia promotes inflammation, while in some instances, such as in tumor sites, low oxygen levels generally cause unresponsiveness of immune effectors, thus favoring tumor growth. The often-opposing effects displayed by HIF activation on the activity of immune cells equally match this complexity (*Corrado and Fontana, 2020*). Previous work has indeed highlighted the important role of the PHD-HIF axis in regulating both innate and adaptive immune effectors (*Watts and Walmsley, 2019*). The role of HIF1α in regulating T cell activity has been described in many studies and is mainly linked to the capacity of this hypoxia-induced transcription factor to promote glycolysis (for a review, see *McGettrick and O'Neill, 2020*). Accordingly, HIF1α expression favors the development of highly glycolytic inflammatory Th17 cells, while inhibiting the development of Tregs, which rely mostly on aerobic metabolism (*Shi et al., 2011*). HIF1α also plays a direct role in Th17 development, through the transcriptional activation of *Rorc* (*Dang et al., 2011*). The role of HIF1α in Th1 development appears as more complex and context-dependent. Hypoxia decreases IFN-γ production of Th1-like cells in a HIF1α-dependent fashion (*Shehade et al., 2015*), while sustained expression of this transcription factor in normoxia (as observed in mice lacking PHDs expression; *Clever et al., 2016*) leads to an increase in IFN-γ-secreting CD4$^+$ T cells. Of note, expression of HIF1α can be upregulated in normoxia both by TCR (*Lukashev et al., 2001*) and cytokine-initiated signals (*Dang et al., 2011*), confirming that HIF1α may play a role in Th1 cells development both in hypoxia and normoxia.

The role of hypoxia-induced factors in Treg development and function is presently not fully elucidated. As previously discussed, HIF1α deficiency improves Treg cell development, possibly a consequence of the limited requirement for glycolysis of this cell subset (*Shi et al., 2011*). However, hypoxia promotes Foxp3 expression in a HIF1α-dependent fashion (*Ben-Shoshan et al., 2008*) and expression of HIF1α is required for adequate regulatory T cell function (*Clambey et al., 2012*). Similarly, a recent report has identified HIF2α as an important mediator of Treg function in vivo, further stressing the important role of these hypoxia-induced factors in the control of in vivo inflammatory manifestations (*Hsu et al., 2020*). In agreement with these conclusions, PHD proteins have also been shown to play a role in the differentiation of peripheral (but not thymic-derived) Tregs (*Clever et al., 2016*). Expression of these proteins appears to redundantly regulate Th1 vs. iTreg development, mostly by limiting the accumulation of HIF1α. In contrast to this study, a recent publication has highlighted a selective role of the PHD2 isoform in the regulation of Treg function (*Yamamoto et al., 2019*). ShRNA-mediated knockdown of PHD2 expression in Foxp3-expressing cells (PHD2-KD Tregs) led to a systemic inflammatory syndrome characterized by mononuclear cell infiltration in several organs. PHD2-KD Tregs displayed reduced suppressive capacities both in vitro and in vivo, suggesting an important and intrinsic role of PHD2 in this cell subset. Of interest, loss of HIF2α expression reversed the phenotype of these mice bearing PHD2-KD Tregs, suggesting an important role of the PHD2-HIF2α axis in regulating Treg function.

To better delineate the role of PHD2, HIF1α, and HIF2α in the regulation of Treg development and function, we have generated a set of conditional mouse strains lacking expression of these hypoxia-responsive proteins in Foxp3$^+$ cells. Using these tools, we confirm herein that mice in which expression of PHD2 is selectively inactivated in regulatory T cells display a spontaneous inflammatory syndrome characterized by altered immune homeostasis at the steady state and high sensitivity to Th1-type inflammatory diseases. This proinflammatory phenotype was accentuated by the concomitant loss of PHD2 and HIF1α, but almost completely alleviated in mice dually deficient for PHD2 and HIF2α. Transcriptome analysis confirmed a marginal role for HIF1α-dependent enhanced glycolysis in the regulation of Treg function and allowed us to identify STAT1 as a potential target of the PHD2-HIF2α axis in maintaining immune homeostasis and preventing excessive Th1-mediated inflammation.

## Results

## Deletion of PHD2 in Tregs leads to a systemic, type 1-like, inflammatory syndrome associated to altered Treg numbers and phenotype

Based on the predominance of *Egln1* (PHD2) expression in Tregs over other members of the PHD family, we generated a mouse strain lacking PHD2 expression in Tregs (identified as PHD2$^{\Delta Treg}$) (*Figure 1*, *Figure 1—figure supplement 1*), as described in Materials and methods. These mice displayed a strongly reduced expression of *Egln1* mRNA in Tregs, while retaining control level expression of this enzyme in other, non-Treg spleen and peripheral lymph node cells (*Figure 1— figure supplement 1*). Upregulation of GLUT1 expression, a well-known target of HIF1α, was also only found in Foxp3-expressing cells in these mice, further supporting the selective depletion of PHD2 in Tregs vs. T convs (*Figure 1—figure supplement 1*). While fertile and viable, over 70% of these mice developed a spontaneous inflammatory syndrome, characterized by weight loss, episodes of anal prolapse, reduced colon length, splenomegaly, and hemorrhagic abdomen (*Figure 1a–e*). This last feature is most likely due to an increased blood hematocrit (with enhanced numbers of circulating red blood cells) associated to an elevation in vascular permeability, as shown in *Figure 1—figure supplement 2*. Although the frequency of CD4$^+$ and CD8$^+$ conventional T lymphocytes (see *Figure 1—figure supplement 3* for gating strategy) in several lymphoid organs was not significantly altered in PHD2$^{\Delta Treg}$ mice (*Figure 1f*), the total number of CD4$^+$ cells was increased in the peripheral lymphoid organs of these mice (*Figure 1—figure supplement 4*). Moreover, these lymphocytes displayed clear signs of spontaneous activation, as evidenced by the significant increase in the expression of markers (i.e., CD44) associated to an effector-like phenotype (*Figure 1g and h*). Confirming these findings, intracellular staining of short-term stimulated T cells (using pharmacological agents bypassing TCR signaling) revealed an increased capacity of conventional T cells from PHD2$^{\Delta Treg}$ mice to produce IFN-γ, while retaining control-like production of IL-17 (*Figure 1i*). The ex vivo evaluation of mRNA abundance in whole, unfractionated, mesenteric lymph nodes (mLNs) similarly showed a significantly elevated expression of type 1-associated adaptive and innate cytokines, including *Ifng*, *Il1b,* both *Il12* subunits, and *Tnfa* (*Figure 1j*). Overall, these observations point to the establishment of a Th1-like, proinflammatory environment in mice possessing PHD2-deficient Tregs.

Much to our surprise, flow cytometric analysis of lymphoid organs from naive animals revealed an increased frequency of Treg cells in the spleen, lymph nodes, and lamina propria of PHD2$^{\Delta Treg}$ mice when compared to control animals (*Figure 2a*). To evaluate the possible influence of PHD2 deletion on Treg development, thymic cell suspensions were analyzed for the expression of early Treg markers, including Foxp3, CD25, and CD24 (*Owen et al., 2019*). Recent studies have revealed that mature Foxp3$^{high}$ CD25$^+$ Tregs can differentiate from two distinct thymic precursors identified as respectively CD25$^+$ Foxp3$^-$ and CD25$^-$ Foxp3$^{low}$ precursor Tregs (pre-Tregs). Analysis of thymic cell suspensions revealed an accumulation of the Foxp3$^{low}$ pre-Tregs and a reduction in the number of mature Tregs in PHD2-deficient, Foxp3-expressing cells, suggesting an early role for PHD2 in the generation of thymic-derived Tregs (*Figure 2b and c*). Accordingly, PHD2-deficient, Foxp3-expressing cells retained higher expression of CD24 (*Figure 2d*), a marker associated to a thymic immature state (*Owen et al., 2019*), further confirming a putative role for PHD2 in the development of thymic-derived Tregs. No difference in the relative frequency of Treg subsets identified by the co-expression of Foxp3 with either naive and memory markers (*Figure 2e*) or with master transcription factors T-bet, GATA3, or RORγt (*Figure 2f*) was noted in these mice. The phenotype of splenic, PHD2-deficient Tregs was however significantly altered, showing a slight, but statistically significant, reduction in the expression of Foxp3 (*Figure 2g*), accompanied by reduced expression of the CD25, ICOS, and CD44 markers and enhanced expression of PD-1 (*Figure 2h*). Of note, neither CTLA-4 (*Figure 2h*) nor *Il10* (*Figure 2i*) expression was altered in PHD2-deficient Tregs. To evaluate the functional consequences of PHD2 deletion on peripheral Treg development, we generated Tregs from naive, conventional T cells using a well-established in vitro protocol. In keeping with in vivo observations, culture of CD4$^+$ T conv from PHD2$^{\Delta Treg}$ mice led to a consistently higher yield of Foxp3-expressing cells when activated in the presence of Treg-inducing cytokines (*Figure 2j and k*). In contrast to their in vivo counterparts, these induced Tregs displayed control level expression of Foxp3 (*Figure 2l*).

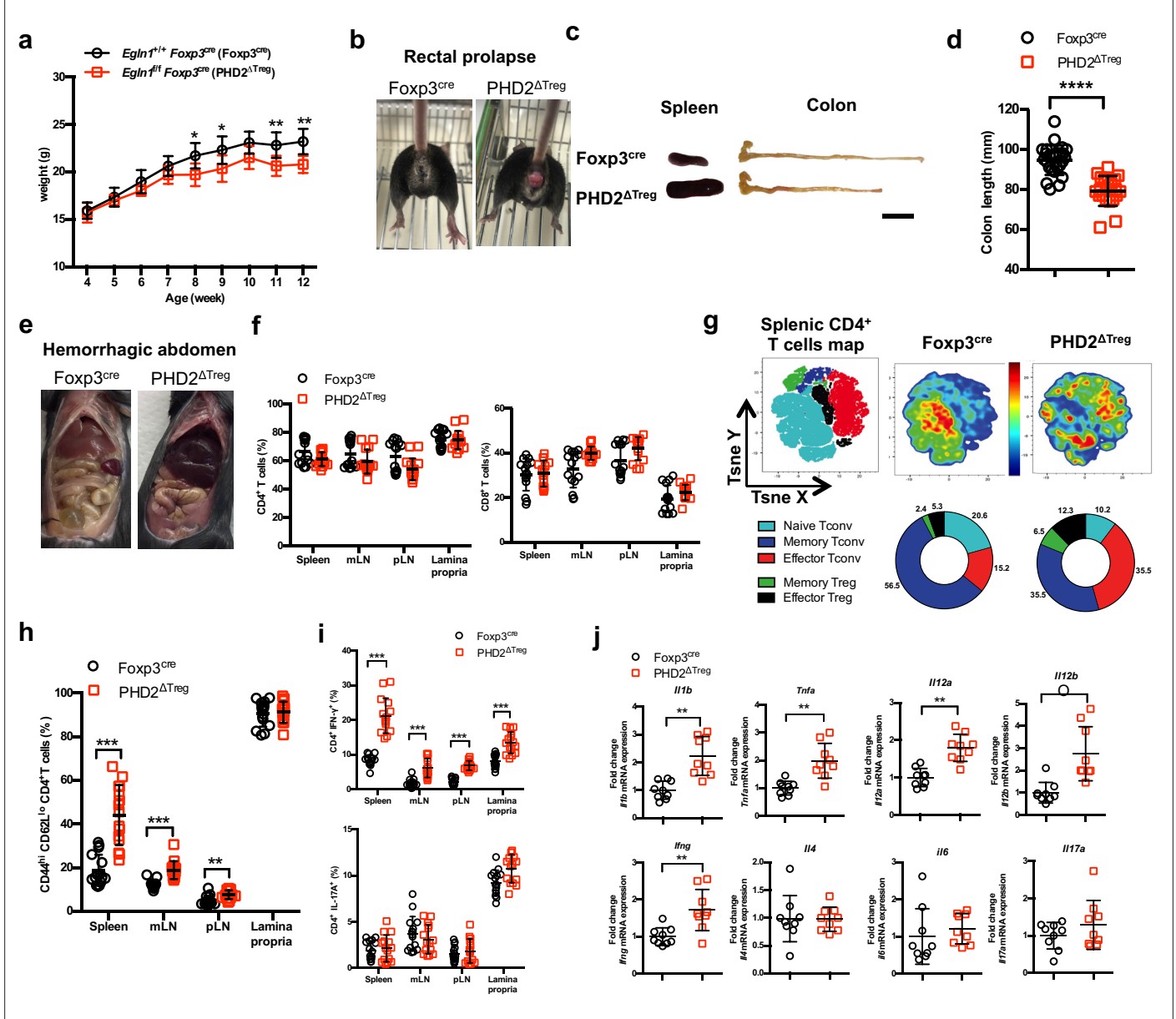

**Figure 1.** PHD2[ΔTreg] mice display a spontaneous Th1-like inflammatory syndrome. (**a**) Body weight of naive mice was determined weekly. (**b**) At 12 weeks of age, male and female mice were examined for rectal prolapse. (**c**) Splenomegaly and colon length summarized in (**d**). (**e**) Representative gross autopsy of a hemorrhagic abdomen, (**f–i**) Lymphoid cells from spleen, mesenteric (mLN), peripheral (pLN) lymph nodes, or the small intestine lamina propria were collected from Foxp3[cre] and PHD2[ΔTreg] mice. (**f**) Frequency of conventional, Foxp3- CD4 and CD8-expressing cells among TCRβ-expressing T lymphocytes. (**g**) Representative merged (n = 15) t-distributed stochastic neighbor embedding (t-SNE) plot after dimensionality reduction and unsupervised clustering of flow cytometry data from CD4-expressing spleen cells. Relative distributions of CD4[+] lymphocyte subsets are shown as doughnut charts. (**h**) Frequency of effector-like (CD44[hi] CD62L[lo]) conventional T lymphocytes in the indicated lymphoid organs. (**i**) Frequency of IFN-γ (top panel) and IL-17A (bottom panel) producing CD4[+] T cells after in vitro stimulation. (**j**) Expression of inflammatory cytokines determined by qPCR on extracts from unfractionated mLNs. Data are representative of at least three independent experiments with n = 9 (**a, j**), n = 25 (**d**), and n = 15 (**f–i**) per group. Values are presented as the mean ± SD and were compared by two-tailed unpaired Student's *t*-test. Only significant differences are indicated as follows: *p<0.05, **p<0.01, ***p<0.001, ****p<0.0001. Naive Tconv: Foxp3[-] CD44[-] CD62L[+]; Memory Tconv: Foxp3[-] CD44[+] CD62L[+]; Effector Tconv: Foxp3[-] CD44[+] CD62L[-]; Memory Treg: Foxp3[+] CD44[+]CD62L[+]; Effector Treg: Foxp3[+] CD44[+]CD62L[-].

The online version of this article includes the following source data and figure supplement(s) for figure 1:

**Source data 1.** PHD2[ΔTreg] mice display a spontaneous Th1-like inflammatory syndrome.

**Figure supplement 1.** Treg-restricted loss of *Egln1* gene expression in PHD2[ΔTreg] mice.

**Figure supplement 1—source data 1.** Treg-restricted loss of *Egln1* gene expression in PHD2[ΔTreg] mice.

*Figure 1 continued on next page*

Figure 1 continued

**Figure supplement 2.** Increased blood cells counts and elevated hematocrit in PHD2$^{\Delta Treg}$ mice associated with an increase in vascular permeability.

**Figure supplement 2—source data 1.** Increased blood cells counts and elevated hematocrit in PHD2$^{\Delta Treg}$ mice associated with an increase in vascular permeability.

**Figure supplement 3.** Gating strategy for flow cytometry data analysis.

**Figure supplement 4.** Absolute cell counts.

**Figure supplement 4—source data 1.** Absolute cell counts.

To evaluate whether the altered phenotype of PHD2-deficient Tregs was a cell-autonomous phenomenon, heterozygous *Foxp3*$^{cre/+}$ *Egln1*$^{fl/fl}$ mice in which both PHD2-sufficient (YFP-negative) and PHD2-deficient (YFP-positive) Tregs coexist were examined (*Figure 3*). These mice did not display any sign of inflammation or hematological dysfunction and were morphologically (cf. weight, colon length, and spleen size) indistinguishable from Foxp3$^{cre}$ or Foxp3$^{cre/+}$ mice (this latter strain displaying the expected 1:1 ratio of YFP-pos:YFP-neg cells). Surprisingly, WT Tregs outcompeted PHD2-deficient Tregs in all compartments examined in *Foxp3*$^{cre/+}$ *Egln1*$^{fl/fl}$ mice (i.e., thymus, spleen, and peripheral lymph nodes, *Figure 3a*). A similar trend was observed following the transfer of an equal mix of WT and PHD2-deficient Tregs in *Rag2*-deficient mice (data not shown), strongly suggesting that PHD2 expression plays a role in Treg fitness and survival in the periphery. As previously shown in *Figure 2*, PHD2-deficient Tregs expressed lower levels of Foxp3, CD25, and CD44, indicative of an intrinsic role of PHD2 in regulating Treg phenotype (*Figure 3b–e*). However, expression of CTLA-4 was not altered in PHD2-deficient Tregs (*Figure 3b and f*). Whether the altered fitness/ capacity to repopulate the periphery of PHD2-deficient Tregs is due to reduced expression of CD25 remains to be established.

## In vivo-reduced suppressive function of PHD2-deficient Tregs

To evaluate the suppressive capacity of PHD2-deficient Tregs cells, ex vivo-purified CD45.2-expressing Tregs from control and PHD2$^{\Delta Treg}$ mice were adoptively co-transferred into syngeneic Rag-deficient mice with CFSE-labeled, CD45.1-expressing CD4$^+$ naive T lymphocytes (*Figure 4a*). In the absence of Tregs, transferred naive cells rapidly divided and acquired an effector-like phenotype, a well-established consequence of homeostatic proliferation in a lymphopenic environment (*Figure 4b*). Addition of WT Tregs in the inoculum led to a significant reduction of conventional T cell proliferation and phenotype switch, while PHD2-deficient Tregs appeared functionally impaired in this assay (*Figure 4b–d*). Lack of suppressive activity of these Tregs was not a consequence of reduced viability and/or in vivo survival, as shown by the normal recovery rate of both Treg-populations at the time of assay readout (*Figure 4e*). In contrast, when tested in vitro, PHD2-deficient Tregs consistently displayed a fully functional suppressive activity (*Figure 4f and g*).

## Increased susceptibility of PHD2$^{\Delta Treg}$ mice to type 1 experimental inflammation

A series of experimental acute and chronic inflammatory models were employed to further evaluate the capacity of PHD2$^{\Delta Treg}$ mice to sustain an in vivo inflammatory challenge. We first exposed mice to a chemical-induced colitis protocol. This assay revealed an increased sensitivity of PHD2$^{\Delta Treg}$ mice to most dextran sodium sulfate (DSS)-induced inflammatory manifestations, including weight loss (*Figure 5a*), survival (*Figure 5b*), clinical score (*Figure 5c*), and colon length (*Figure 5d*). No difference was noted, however, in crypt morphology induced by DSS in both mouse strains (*Figure 5e*). Similar observations were made when mice were acutely infected with *Toxoplasma gondii*, a model of infection-induced pathology (*Figure 5f*). Infected PHD2$^{\Delta Treg}$ mice displayed increased weight loss (*Figure 5g*), reduced colon length (*Figure 5h*), and increased frequency of activated cells characterized by an effector-like phenotype (*Figure 5i*) and IFN-γ secretion capacity (*Figure 5j*). Infected PHD2$^{\Delta Treg}$ mice also displayed a decrease in Treg T-bet$^+$ frequency, a population known to control Th1 inflammation during toxoplasmosis (*Wohlfert et al., 2020*; *Figure 5k*). Overall, PHD2$^{\Delta Treg}$ mice displayed an uncontrolled expansion of Th1-like cells following experimental toxoplasmosis. In contrast, both PHD2-deficient and -sufficient mouse strains were equally sensitive to enteritis induced upon injection of anti-CD3 antibodies (*Figure 5—figure supplement 1*), a model known to induce the predominant expansion of Th17-like, inflammatory effectors in vivo (*Esplugues et al., 2011*). The role of uncontrolled IFN-γ secretion in

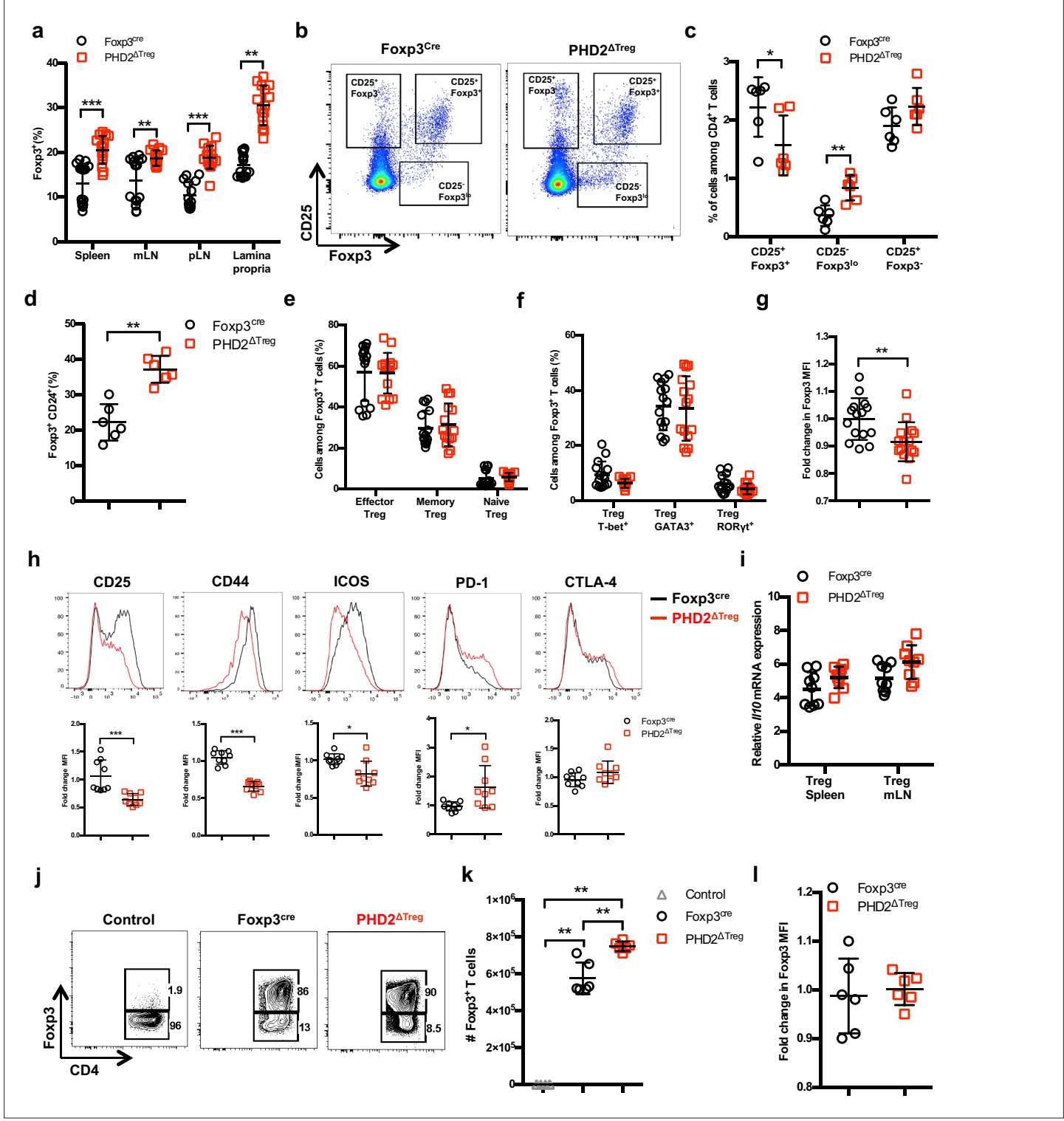

**Figure 2.** Increased number, but altered phenotype of PHD2-deficient Treg cells. Lymphoid cells from the thymus, spleen, mesenteric (mLN), and peripheral (pLN) lymph nodes were collected at 12 weeks of age from Foxp3cre and PHD2ΔTreg male and female mice, and the relative frequency and phenotype of Foxp3-expressing cells were established by flow cytometry or qPCR. (**a**) Frequency of Foxp3-expressing cells among CD4-positive T lymphocytes. (**b**) Representative flow cytometry expression profiles of Foxp3 and CD25 expression among thymic CD4+ T cells. (**c**) Frequency of mature-like (CD25+ Foxp3+) and Treg precursors subsets identified respectively as CD25- Foxp3lo and CD25+ Foxp3- cells among thymic CD4+ T cells. (**d**) Frequency of immature-like, CD24+ Foxp3+ T cells in the thymus of adult mice. (**e**) Frequency of effector (CD62Llow CD44high), memory (CD62Lhigh CD44high), and naive (CD62Lhigh CD44low) splenic Foxp3-expressing cells. (**f**) Frequency of splenic Tregs expressing the master transcription factors T-bet,

*Figure 2 continued on next page*

*Figure 2 continued*

GATA3, and RORγt. (**g**) Ratio of the Foxp3 MFI of PHD2-KO splenic Tregs to Foxp3[cre] splenic Tregs. (**h**) Expression of CD25, CD44, ICOS, PD-1, and CTLA-4 in splenic Treg of Foxp3[cre] and PHD2[ΔTreg] mice. Top panel: representative traces of MFI. Bottom panel: ratios of the MFIs of PHD2-KO Treg to Foxp3[cre] Treg cells are expressed as the mean ± SD. (**i**) *Il10* gene expression relative to RPL32 by ex vivo-purified Tregs was determined by qPCR. (**j–l**) CD4[+] Foxp3[-] splenic naive T cells were stimulated in vitro with anti-CD3/CD28 (5/1 μg/ml) in the presence of TGF-β (3 μg/ml) and IL-2 (10 μg/ml) for 72 hr to induce Treg polarization. (**j**) Representative flow cytometry expression profiles of Foxp3 expression at the end of the culture period. The first panel represents a typical profile of cells activated in the absence of polarizing cytokines. (**k**) Number of Foxp3[+] cells generated in the culture conditions. (**l**) Expression (MFI) of Foxp3 by in vitro-induced Treg cells. Data are representative of at least two independent experiments with n = 15 (**a, e–g**) n = 9 (**h, i**) or n = 6 (**c, d, k, l**) per group. Values are presented as the mean ± SD and were compared by two-tailed unpaired Student's *t*-test. Only significant differences are indicated as follows: *p<0.05, **p<0.01, ***p<0.001.

The online version of this article includes the following source data for figure 2:

**Source data 1.** Increased number, but altered phenotype of PHD2-deficient Treg cells.

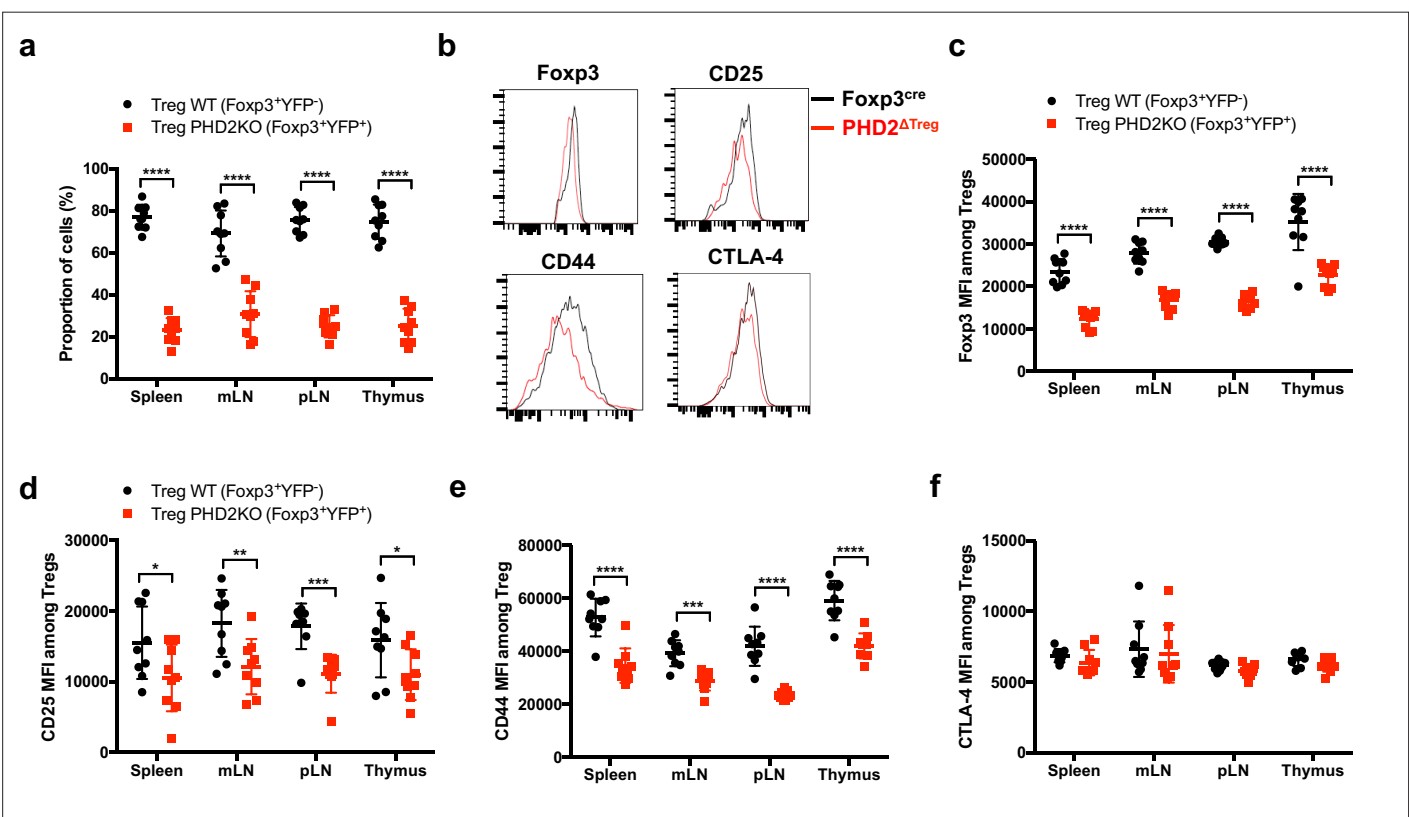

**Figure 3.** Cell-autonomous role of PHD2 in determining Treg cells phenotype. Spleen, thymus, mesenteric (mLN), and peripheral (pLN) lymph nodes were collected at 8 weeks of age from *Foxp3*[cre/+]*Egln1*[f/f] heterozygous female mice, and the relative frequency and phenotype of Foxp3-expressing cells were established by flow cytometry. (**a**) Proportion of WT (YFP- cells) or PHD2-KO (YFP+ cells) Treg cells among Foxp3-expressing cells. (**b**) Representative histograms of Foxp3, CD25, CD44, and CTLA-4 expression in splenic WT Tregs (black lines) compared to splenic PHD2-KO Tregs (red lines). (**c**) Foxp3 MFI, (**d**) CD25 MFI, (**e**) CD44 MFI, and (**f**) CTLA-4 MFI of WT and PHD2-KO Tregs in lymphoid organs. Data are representative of two independent experiments with n = 9 per group. Values are presented as the mean ± SD and were compared by two-tailed unpaired Student's *t*-test. Only significant differences are indicated as follows: *p<0.05, **p<0.01, ***p<0.001, ****p<0.0001.

The online version of this article includes the following source data for figure 3:

**Source data 1.** Cell-autonomous role of PHD2 in determining Treg cells phenotype.

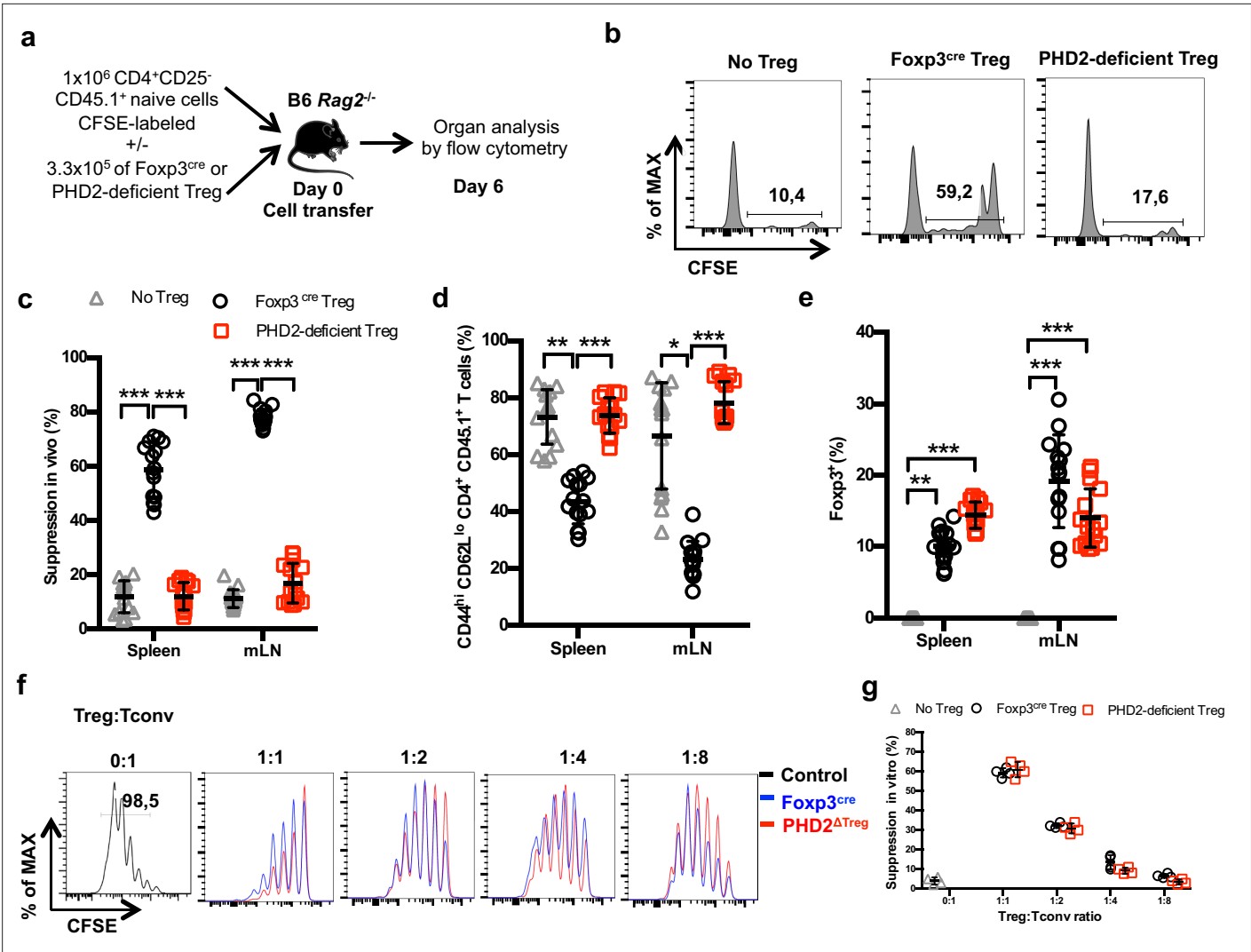

**Figure 4.** Reduced in vivo but not in vitro suppressive capacity of PHD2-deficient Treg. (**a**) Treg function was assayed following adoptive co-transfer of CD45.2 Foxp3-expressing cells with naive, CFSE-labeled congenic CD45.1 CD4$^+$ lymphocytes (Treg: Tconv ratio 1:3) into syngeneic lymphopenic male mice (*Rag2$^{-/-}$*). Recipient mice were euthanized at day 6 post-transfer, and their spleen and mesenteric lymph node (mLN) cells analyzed by flow cytometry. (**b**) Representative flow cytometry expression profiles of CFSE-labeled cells (CD45.1 gate in the spleen) with or without co-transferred Foxp3$^+$ cells from Foxp3$^{cre}$ or PHD2$^{\Delta Treg}$ male mice. (**c**) Percentage of suppression established from CFSE staining profiles. (**d**) Frequency of activated (CD4$^+$ CD45.1$^+$ CD44$^{hi}$ CD62L$^{lo}$) cells in the indicated lymphoid organs. (**e**) Frequency of Treg cells in the indicated organs 6 days post-transfer. (**f, g**) CFSE-labeled, naive conventional CD4$^+$ T cells from CD54.1 mice were co-cultured with ex vivo-purified Treg cells from Foxp3$^{cre}$ or PHD2$^{\Delta Treg}$ mice at the indicated ratios in the presence of anti-CD3 antibodies (0.5 µg/ml) and splenic feeder cells. (**f**) Representative flow cytometry profiles of CSFE staining. (**g**) Percent of suppression of proliferation as compared to cultures in which Treg cells were omitted. Data are representative of three independent experiments with n = 15 (**b–e**) or n = 4 (**f, g**) per group. Values are presented as the mean ± SD and were compared by two-way ANOVA with Tukey's multiple comparisons test (**c–e**) or two-tailed unpaired Student's *t*-test (**g**). Only significant differences are indicated as follows: *p<0.05, **p<0.01, ***p<0.001.

The online version of this article includes the following source data for figure 4:

**Source data 1.** Reduced in vivo but not in vitro suppressive capacity of PHD2-deficient Treg.

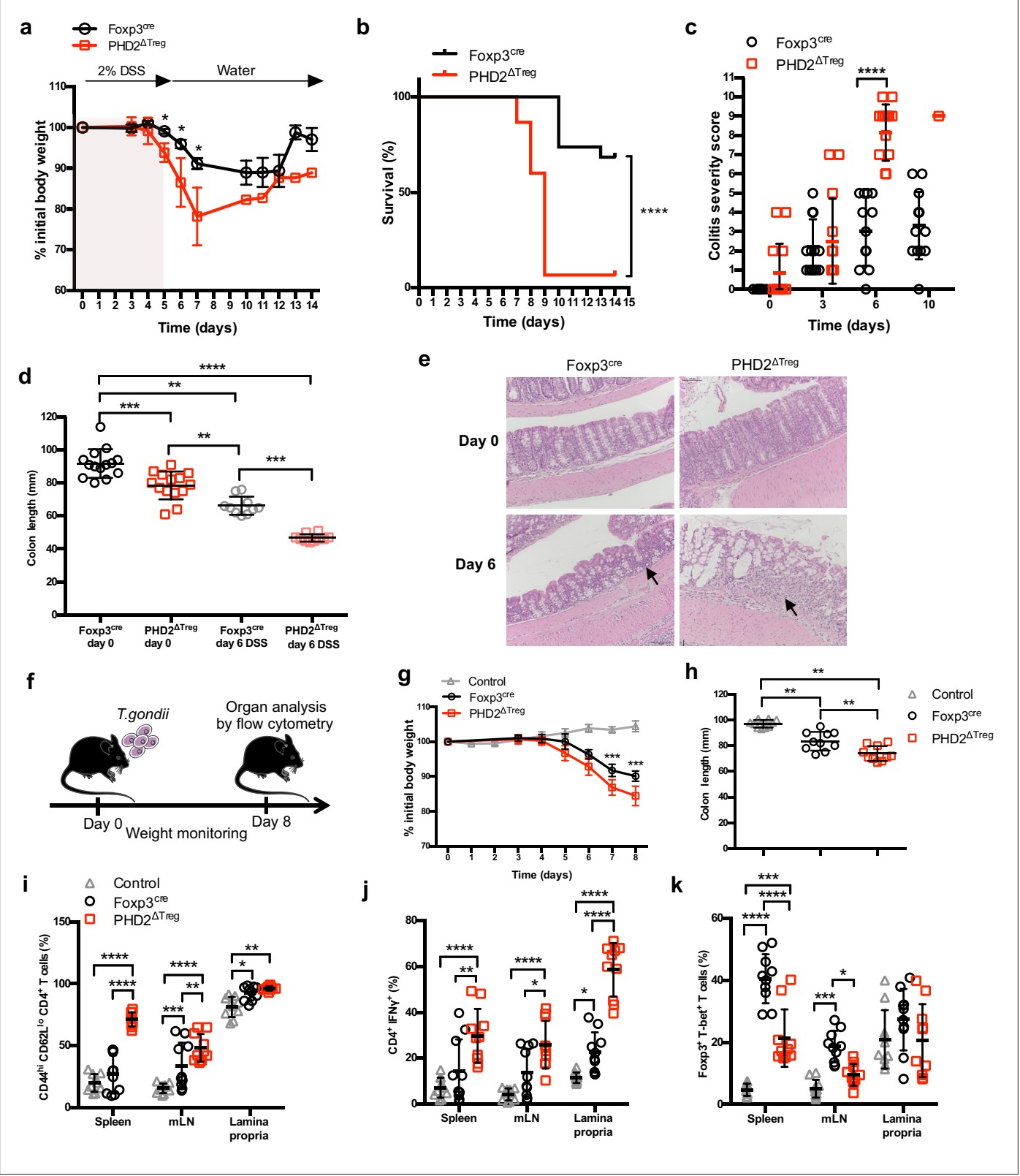

**Figure 5.** Increased sensitivity of PHD2$^{\Delta Treg}$ mice to dextran sodium sulfate (DSS)-induced colitis and toxoplasmosis. Foxp3$^{cre}$ and PHD2$^{\Delta Treg}$ male mice were provided with 2% DSS in tap water for 5 days. On day 5, the 2% DSS water was replaced with normal drinking water and mice were followed during 14 days for (**a**) body weight, (**b**) survival, (**c**) colitis severity, and (**d**) colon length. (**e**) Colons were isolated from untreated mice or 6 days after colitis induction and were fixed and stained with hematoxylin and eosin (H&E); arrows indicate inflammatory cell infiltrates. (**f**) Foxp3$^{cre}$ and PHD2$^{\Delta Treg}$

*Figure 5 continued*

male mice were infected by intragastric gavage with 10 cysts of ME-49 type II *Toxoplasma gondii* (control group are Foxp3$^{cre}$ mice without treatment) and subsequently followed for (**g**) body weight. (**h**) Mice were sacrificed 8 days after infection to assess colon length. (**i**) Frequency of effector-like (CD44$^{hi}$ CD62L$^{lo}$) conventional T lymphocytes in the indicated lymphoid organs. (**j**) Frequency of IFN-γ-producing CD4$^+$ T cells after in vitro stimulation. (**k**) Frequency of T-bet$^+$ among Foxp3$^+$(Treg) cells. Data are representative of three independent experiments with n = 20 (**a, b**), n = 10–14 (**c, d**), n = 5 (**e**), or n = 10 (**g–k**) per group. Values are presented as the mean ± SD and were compared by two-tailed unpaired Student's *t*-test (**a, c, g**), by Mantel–Cox test (**b**), one-way ANOVA with Tukey's multiple comparisons test (**d, h**) or two-way ANOVA with Tukey's multiple comparisons test (**i–k**). Only significant differences are indicated as follows: *$p<0.05$, **$p<0.01$, ***$p<0.001$, ****$p<0.0001$.

The online version of this article includes the following source data and figure supplement(s) for figure 5:

**Source data 1.** Increased sensitivity of PHD2$^{ΔTreg}$ mice to dextran sodium sulfate (DSS)-induced colitis and toxoplasmosis.

**Figure supplement 1.** PHD2$^{ΔTreg}$ mice display a near-normal response to anti-CD3-induced enteritis.

**Figure supplement 1—source data 1.** PHD2$^{ΔTreg}$ mice display a near-normal response to anti-CD3-induced enteritis.

**Figure supplement 2.** Loss of *Ifng* gene expression attenuates the proinflammatory phenotype of PHD2$^{ΔTreg}$ mice.

**Figure supplement 2—source data 1.** Loss of *Ifng* gene expression attenuates the proinflammatory phenotype of PHD2$^{ΔTreg}$ mice.

mediating the proinflammatory status of this mouse strain was further confirmed by the observation that ubiquitous loss of *Ifng* gene expression largely reversed the phenotypical and cellular altered status of PHD2$^{ΔTreg}$ mice (***Figure 5—figure supplement 2***).

## Concomitant loss of HIF2α, but not HIF1α, expression partially corrects the proinflammatory phenotype of PHD2$^{ΔTreg}$ mice

Based on the notion that HIF1α and HIF2α represent well-described targets of PHD2, we established a series of conditional KOs mouse strains to identify the molecular pathway responsible for the decreased functional activity of PHD2-deficient Tregs at steady state (***Figure 6***; see ***Figure 6—figure supplement 1*** for strain validation). Treg-selective deletion of HIF1α and HIF2α expression alone did not significantly alter colon length (used as a proxy for spontaneous inflammation) nor general T cell immune homeostasis (***Figure 6—figure supplement 1***). The same observation was made for double HIF1α and HIF2α KOs (data not shown). In marked contrast, combined deletion of PHD2 and HIF2α reversed some of the inflammatory symptoms observed in PHD2$^{ΔTreg}$ mice, such as splenomegaly, colon length (***Figure 6a and b***), and hematocrit counts (***Figure 1—figure supplement 2***). Treg-specific, PHD2-HIF1α double KOs were virtually indistinguishable from PHD2$^{ΔTreg}$ according to these morphological criteria. Noteworthy, however, Treg-specific PHD2-HIF1α double KOs mice were born at sub-Mendelian ratios and displayed a marked weight loss during adult life and reduced viability, indicative of a more pronounced proinflammatory status (data not shown). This mouse strain also displayed a tendency toward increased expansion of Th1-like cells in peripheral lymph nodes (***Figure 6e***). PHD2-HIF1α-HIF2α triple KOs and PHD2-HIF2α double KOs displayed a similar phenotype, establishing a predominant role for HIF2α over HIF1α in mediating the effects of PHD2 on the capacity of Treg to regulate immune homeostasis at rest. Similarly, lack of HIF2α expression largely reversed the altered phenotype of conventional T cells induced by loss of Treg-associated PHD2 expression. Indeed, cells from double (PHD2-HIF2α) and triple (PHD2-HIF1α-HIF2α) Treg-specific KOs displayed a near-normal phenotype (based on CD62L and CD44 expression) and propensity to secrete IFN-γ (***Figure 6c–e***). Finally, loss of Treg-associated expression of HIF2α also reversed the expansion of Treg numbers (***Figure 6f***) and restored Foxp3 protein expression to near-control levels (***Figure 6g***).

## Transcriptomic analysis identifies cell survival, response to chemokines, and STAT1-mediated signaling as target pathways of the PHD2-HIF2α axis in Tregs

Collectively, the previous observations suggest that the PHD2-HIF2α regulatory axis confers to Tregs the capacity to control the spontaneous, type 1-like, activity of conventional T cells. To identify PHD2-dependent signaling pathways operating in Tregs, splenic Foxp3-expressing cells were purified from all mouse strains described in this article and their transcriptome analyzed following bulk RNA-seq. A set of 532 genes were found differentially expressed between WT and PHD2-deficient Tregs (***Figure 7***) (a summary list of upregulated and downregulated pathways in PHD2$^{ΔTreg}$ mice vs. Foxp3$^{cre}$ mice is provided in ***Figure 7—figure supplement 1***). Differential gene expression analysis

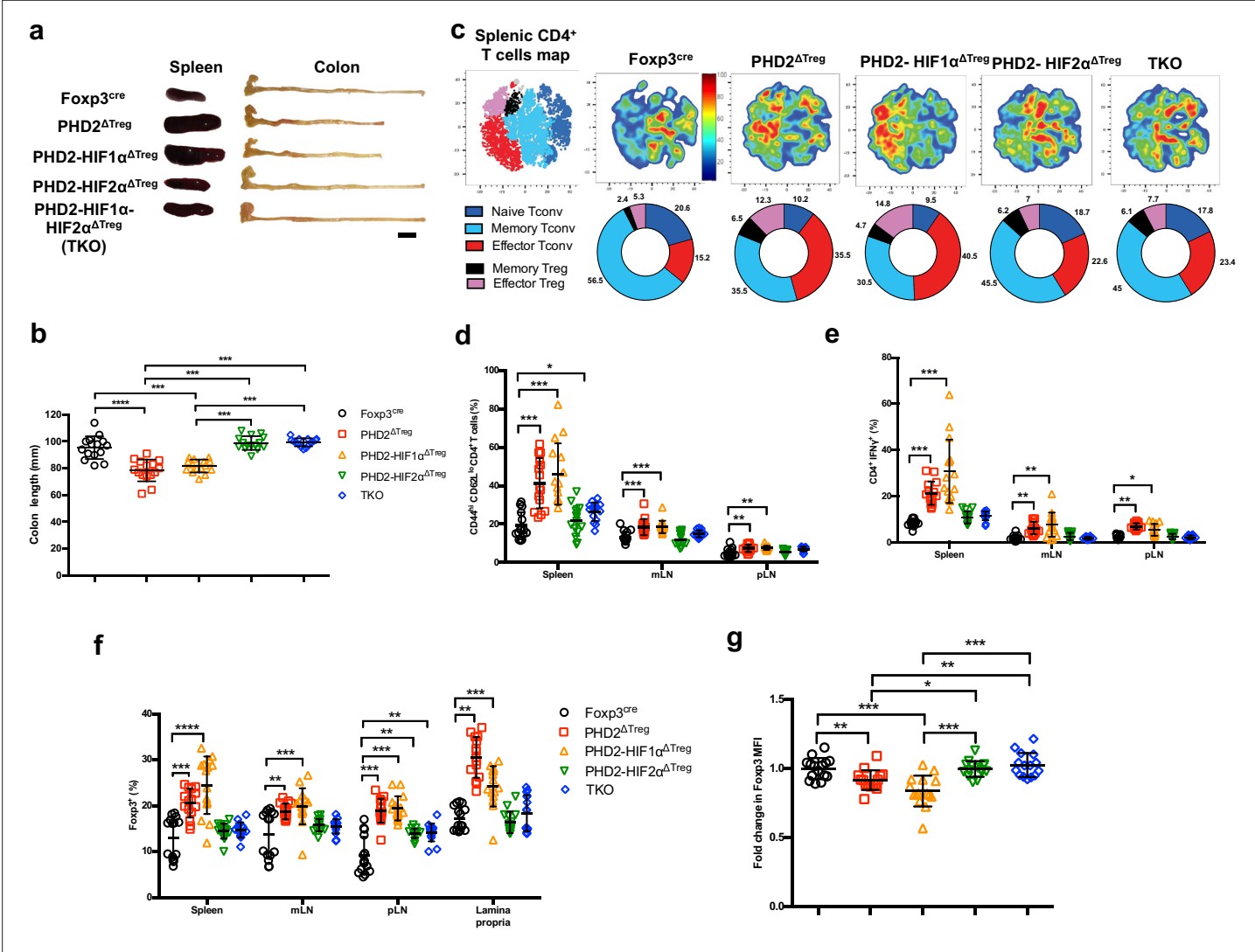

**Figure 6.** Concomitant loss of HIF2α but not HIF1α expression attenuates the proinflammatory phenotype of PHD2^ΔTreg mice. (**a**) Representative gross autopsy of spleens and colon length summarized in (**b**) of Foxp3^cre, PHD2^ΔTreg, PHD2-HIF1α^ΔTreg, PHD2-HIF2α^ΔTreg, and PHD2-HIF1α-HIF2α^ΔTreg (TKO) mice. (**c**) Representative merged (n = 15) t-distributed stochastic neighbor embedding (t-SNE) plot after dimensionality reduction and unsupervised clustering of flow cytometry data from CD4-expressing spleen cells. Relative distributions of CD4⁺ lymphocyte subsets are shown as doughnut charts. (**d–g**) Lymphoid cells from spleen, mesenteric (mLN), peripheral (pLN) lymph nodes, or the small intestine lamina propria were collected from Foxp3^cre, PHD2^ΔTreg, PHD2-HIF1α^ΔTreg mice, PHD2-HIF2α^ΔTreg and PHD2-HIF1α-HIF2α^ΔTreg (TKO) male and female mice and the relative frequency and phenotype of Foxp3-positive and Foxp3-negative, conventional T lymphocytes determined by flow cytometry. (**d**) Frequency of effector-like (CD44^hi CD62L^lo) conventional T lymphocytes in the indicated lymphoid organs. (**e**) Frequency of IFN-γ-producing CD4⁺ T cells after in vitro stimulation. (**f**) Frequency of Foxp3-expressing cells among CD4-positive T lymphocytes. (**g**) Ratio of the Foxp3 MFI of PHD2-KO, PHD2-HIF1αKO, PHD2-HIF2αKO, or TKO splenic Tregs to Foxp3^cre splenic Tregs. Data are representative of at least three independent experiments with n = 15 per groups. Values are expressed as the mean ± SD and were compared by one-way ANOVA with Tukey's multiple comparisons test (**b, g**) or two-way ANOVA with Tukey's multiple comparisons test (**d–f**). Only significant differences are indicated as follows: *p<0.05, **p<0.01, ***p<0.001, ****p<0.0001.

The online version of this article includes the following source data and figure supplement(s) for figure 6:

**Source data 1.** Concomitant loss of HIF2α but not HIF1α expression attenuates the proinflammatory phenotype of PHD2^ΔTreg mice.

**Figure supplement 1.** Treg-selective HIF1α or HIF2α deficiency does not affect immune homeostasis in naive mice.

**Figure supplement 1—source data 1.** Treg-selective HIF1α or HIF2α deficiency does not affect immune homeostasis in naive mice.

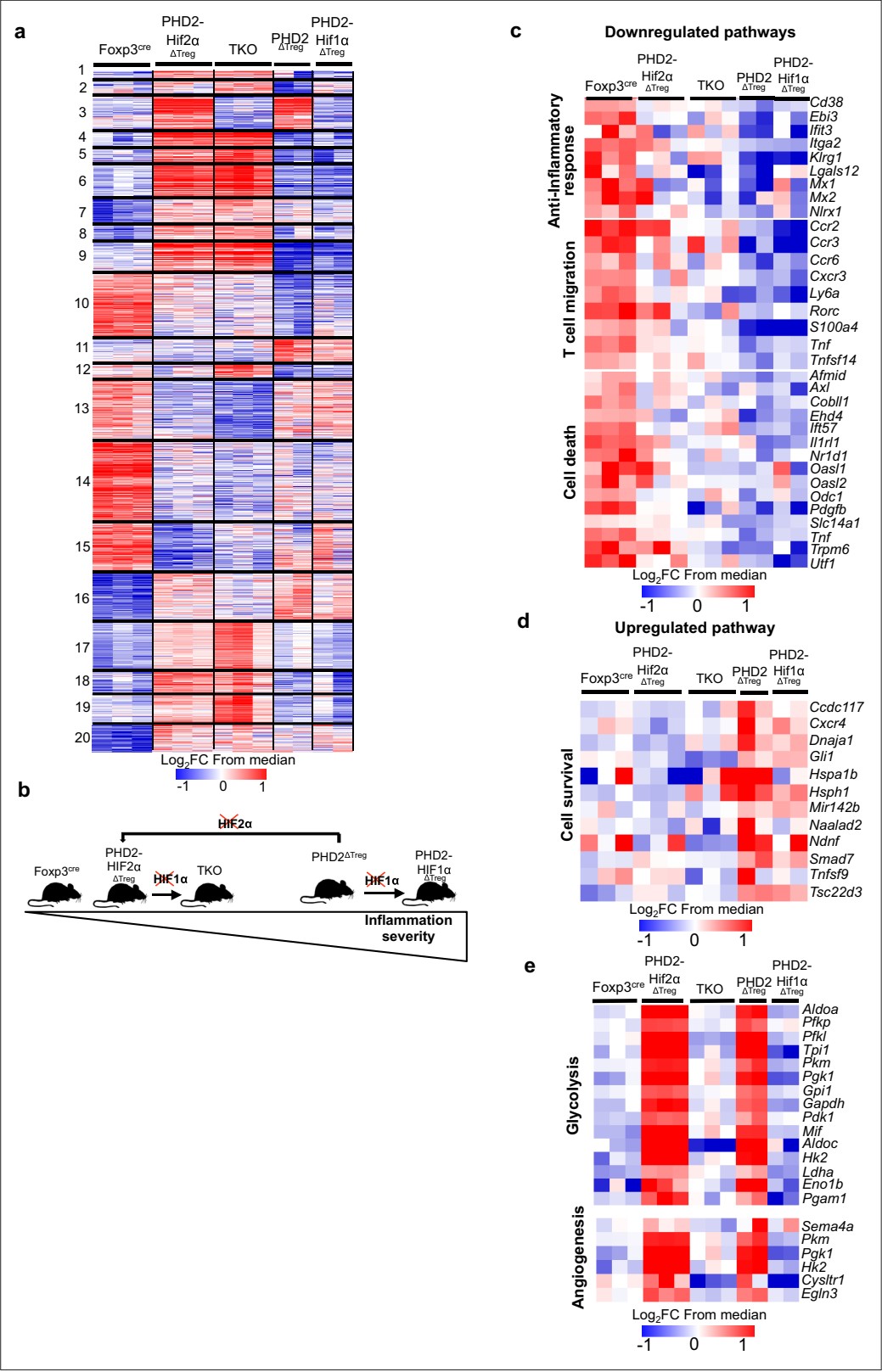

**Figure 7.** Anti-inflammatory response, response to chemokines, and cell survival pathways represent targets of the PHD2-HIF2α axis in Tregs. Splenic Treg cells were purified by cell sorting from Foxp3[cre] (n = 3), PHD2[ΔTreg] (n = 2), PHD2-HIF1α[ΔTreg] (n = 2), PHD2-HIF2α[ΔTreg] (n = 3), and PHD2-HIF1α-HIF2α[ΔTreg] (TKO) (n = 3) male mice, and total RNA was extracted and sequenced by RNA-sequencing (Illumina). (**a**) Heatmap of genes differentially expressed. Values

*Figure 7 continued on next page*

*Figure 7 continued*

are represented as log$_2$ fold change (FC) obtained from median of each gene and are plotted in red-blue color scale with red indicating increased expression and blue indicating decreased expression. Hierarchical clustering of genes (k-means clustering) shows 20 clusters. (**b**) Classification of mouse strains according to their spontaneous inflammation severity. (**c**) Heatmap of genes downregulated when PHD2 and PHD2-HIF1α are deleted and whose expression is restored to a control level (close to Foxp3$^{cre}$ Treg) following deletion of HIF2α (cluster 10, 181 genes). (**d**) Heatmap of genes upregulated when PHD2 and PHD2-HIF1α are deleted and whose expression is restored to a control level following deletion of HIF2α (cluster 11, 66 genes). (**e**) Heatmap of genes upregulated when PHD2 and PHD2-HIF2α are deleted and whose expression is restored to a control level following deletion of HIF1α (cluster 3, 98 genes). Cluster 3, 10, and 11 were subjected to functional annotations and regulatory network analysis in the Ingenuity Pathway Analysis (IPA) software. Data were analyzed using DESeq2, a gene is differentially expressed when log$_2$FC > 0.5 and false discovery rate (FDR) < 0.05.

The online version of this article includes the following source data and figure supplement(s) for figure 7:

**Source data 1.** Anti-inflammatory response, response to chemokines, and cell survival pathways represent targets of the PHD2-HIF2α axis in Tregs.

**Figure supplement 1.** Signaling pathways affected by loss of PHD2 expression in Treg.

**Figure supplement 1—source data 1.** Signaling pathways affected by loss of PHD2 expression in Treg.

between all mouse strains studied identified 1868 genes differentially expressed between groups. An unsupervised clustering of the differentially expressed genes led to the identification of 20 clusters, as shown in *Figure 7a*. To identify the gene clusters that were specifically involved in the immune homeostatic control of naive mice, the RNA-seq data were filtered and grouped by k-means clustering. We next searched for sets of genes whose expression best correlated with an arbitrary inflammatory index, established based on previously described findings (mostly colon length, splenomegaly, and spontaneous conventional T cell activation status) and summarized in *Figure 7b*. In particular, while concomitant deletion of HIF1α expression worsened the inflammatory status of PHD2$^{ΔTreg}$ mice, loss of HIF2α expression mitigated most inflammatory-related parameters at rest. We therefore clustered genes according to a 'gradient of disease severity' and grouped them in sets of gene whose expression decreased (cluster 10, *Figure 7c*) or increased (cluster 11, *Figure 7d*) accordingly. Gene Ontology analysis of these clustered gene sets revealed the following. Reduced expression of cell death-related and gain of survival-associated gene expression correlated with the increased frequency of Tregs in the corresponding mouse strains (*Figure 7c and d*). Not surprisingly, the expression of genes associated with anti-inflammatory responses was gradually lost according to the same severity gradient. Finally, genes, associated with T cell migration, including several chemokine receptors, also displayed an ordered loss of expression along the same gradient (*Figure 7c*). For comparison purposes, genes whose expression was restored to control levels upon combined deletion of PHD2 and HIF1α were also examined. As expected from previously published findings, these HIF1α-dependent biological pathways included glycolysis and angiogenesis (*Figure 7e*).

Ingenuity Pathway Analysis (IPA) was performed in order to identify possible upstream regulators affecting expression of downstream genes identified in clusters 10 and 11. This analysis led to the identification of STAT1 as a putative upstream transcription factor regulating the expression of a set of genes whose expression was altered in PHD2-deficient Tregs (*Figure 8a*). Since *Stat1* mRNA expression was not altered by PHD2 invalidation (as revealed by RNA-seq analysis), we tested the capacity of STAT1 to undergo phosphorylation in response to IFN-γ. This set of experiments led to the identification of a defective, accumulation of phopho-STAT1 in PHD2-deficient Tregs (*Figure 8b and c*), while the levels of total STAT1 protein appeared unaffected (*Figure 8d*). Noteworthy, concomitant deletion of HIF2α restored a near-control response to IFN-γ in PHD2-deficient Tregs (*Figure 8b–d*). In keeping with the observed proinflammatory phenotype associated with these mouse strains, conventional T cells from PHD2$^{ΔTreg}$ mice displayed an augmented response to IFN-γ (as judged by pSTAT1 accumulation), partially reversed in mice bearing Tregs lacking both PHD2 and HIF2α expression (*Figure 8b*, left panel). Finally, the proportion of Tregs expressing CXCR3, a well-described STAT1-dependent chemokine receptor (*Hall et al., 2012*), was reduced in a HIF2α-dependent manner in PHD2-deficient Tregs (*Figure 8e*), further strengthening the conclusion that PHD2 expression controls the response of Tregs to IFN-γ.

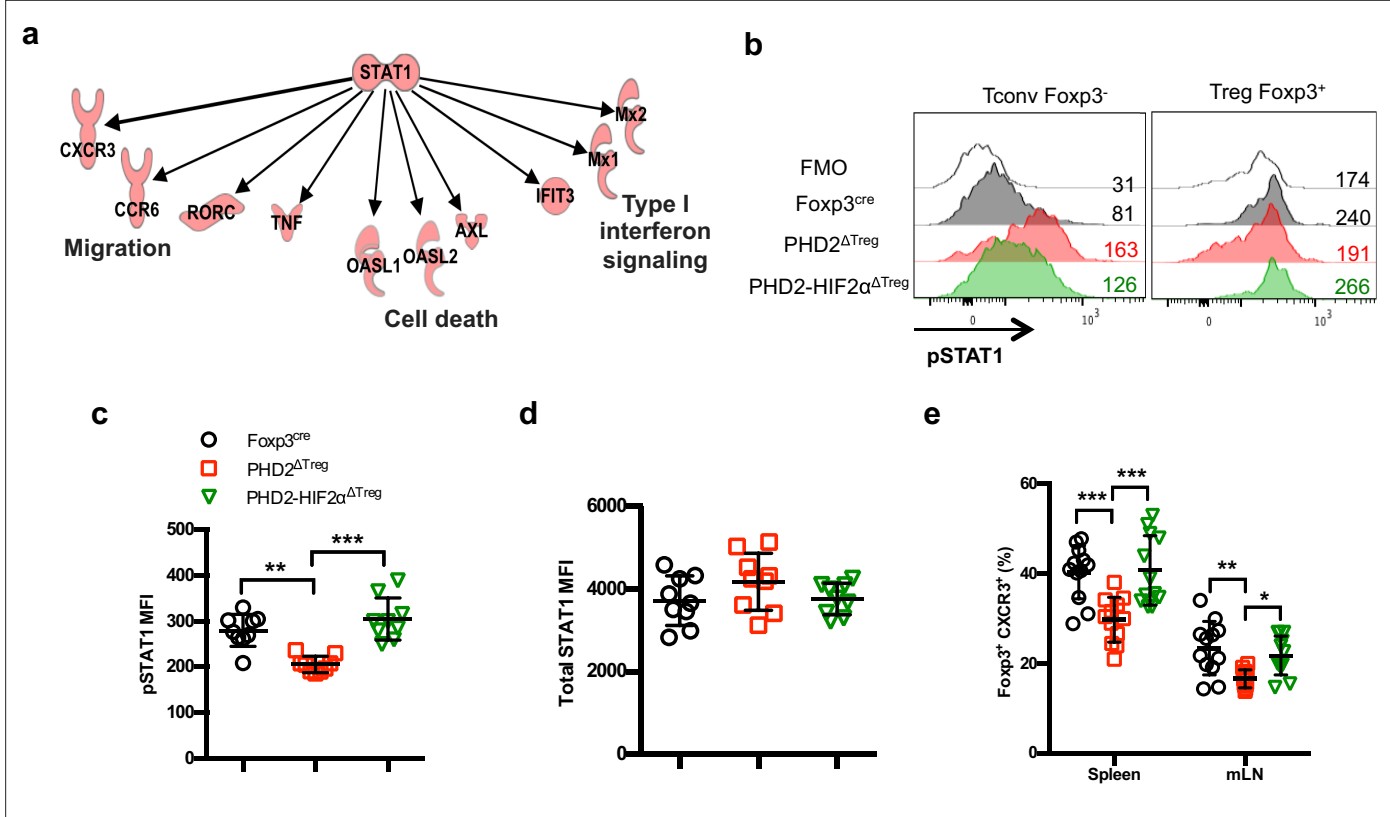

**Figure 8.** Identification of STAT1-mediated signaling as a target of the PHD2-HIF2α axis in Tregs. (**a**) Upregulated and downregulated genes (clusters 10 and 11 in *Figure 6c and d*) were imported into the Ingenuity Pathway Analysis (IPA) software and were subjected to Upstream Regulator Analysis (URA) prediction algorithms. STAT1 was predicted as an upstream regulator of downregulated genes with a p-value=3 × 10$^{-12}$. Phosphorylated form of STAT1 (pSTAT1 [Tyr701]) was assessed by flow cytometry after brief in vitro stimulation (30 min) of splenic CD4$^+$ T lymphocytes with recombinant IFN-γ. (**b**) Representative histogram of pSTAT1 MFI for conventional CD4$^+$ T cells (Tconv) and Treg cells of Foxp3$^{cre}$, PHD2$^{ΔTreg}$, and PHD2-HIF2α$^{ΔTreg}$ male mice. Mean value expression (represented by MFI) of (**c**) pSTAT1 or (**d**) STAT1 total protein by splenic Treg of Foxp3$^{cre}$, PHD2$^{ΔTreg}$, and PHD2-HIF2α$^{ΔTreg}$ mice. (**e**) Frequency of Treg cells expressing the CXCR3 receptor. Data are representative of three independent experiments with n = 9 (**b–d**) or n = 12 (**e**) per groups. Values are presented as the mean ± SD and were compared by one-way ANOVA with Tukey's multiple comparisons. Only significant differences are indicated as follows: **p<0.01, ***p<0.001.

The online version of this article includes the following source data and figure supplement(s) for figure 8:

**Source data 1.** Identification of STAT1-mediated signaling as a target of the PHD2-HIF2α axis in Tregs.

**Figure supplement 1.** Gating strategy for identification of YFP$^+$ cells.

# Discussion

This study highlights the important role of the prolyl-hydroxylase PHD2 in the regulation of Treg development and function. Deletion of PHD2 in developing Tregs led to the accumulation of the subset of Treg precursor characterized by low expression of Foxp3 at the expenses of the mature Foxp3$^+$CD25$^+$ Treg population (*Figure 2c*). PHD2-deficient Tregs were nevertheless found in increased numbers in vivo at steady state (*Figure 2a*), albeit with an altered phenotype. In particular, the expression of molecules known to be associated with optimal suppressive activity (such as Foxp3, ICOS ,and CD25) (*Lu et al., 2017*; *Fontenot et al., 2003*; *Redpath et al., 2013*) was marginally decreased, while the expression of PD-1, a marker associated with altered functional activity of many immune cells including Tregs (*Lowther et al., 2016*; *Tan et al., 2021*), was augmented. Of note, other molecules known to play an important role in Treg function were expressed at optimal levels (cf. CTLA4 and IL-10).

Although we have not specifically addressed the role of PHD2 in thymic vs. peripherally induced Tregs, it is noteworthy that lack of PHD2 expression altered Treg thymic development without any major effect on the generation of iTreg in vitro, suggesting a more pronounced role of PHD2 on

thymic vs. peripherally generated Tregs, although this conclusion should be strengthened by additional studies.

Mice selectively lacking PHD2 expression in Treg displayed a proinflammatory phenotype (with early manifestations of gastrointestinal tract inflammation) associated to an elevated hematocrit, enhanced vascular permeability, and an altered homeostatic profile of splenic conventional T cells. In marked contrast to WT Tregs, PHD2-deficient Tregs express relatively high levels of *Vegfa* transcripts (as established from RNA-seq analysis), a factor known to both increase vascular permeability (*Bates, 2010*) and induce erythropoietin (Epo) production from perivascular stromal cells in several organs (*Greenwald et al., 2019*). Alternatively, since the expression of the YFP-Cre allele was found in a minor proportion of CD45-negative cells in all organs examined (see *Figure 8—figure supplement 1*). PHD2 deletion could result in elevated expression of Epo from a nonhematopoietic source. In any event, analysis of several organs (including spleen, liver, and kidney) from PHD2$^{\Delta Treg}$ mice failed to reveal any increase in *Epo* mRNA accumulation (our own unpublished observations), and further studies will therefore be required to identify the mechanism underlying the observed hematological alterations in PHD2$^{\Delta Treg}$ mice. Several observations, however, strongly indicate that the major proinflammatory phenotype observed in these mice were due to the specific impairment in Treg function consequent to the loss of PHD2 activity. Despite previous reports describing the stochastic activity of the *Foxp3*-Cre-YFP allele in non-Tregs leading to the recombination of some, but not all, alleles (*Franckaert et al., 2015*), we consistently found a control-level expression of PHD2 in all conventional T cell subsets tested (*Figure 1—figure supplement 1*). Secondly, co-transfer of highly purified Tregs with WT naive CD4$^+$ conventional target cells clearly demonstrated an intrinsic role of PHD2 in regulating Treg function. Finally, and this will be discussed further below, mice bearing double PHD2-HIF2α-deficient Tregs recover a near-control phenotype, further excluding a major influence of the genetic background on the observed phenotype. Our observations confirm and further extend the observations from a study published during the completion of our work (*Yamamoto et al., 2019*), indicating a specific nonredundant role for PHD2 in controlling Treg activity in vivo.

Based on the notion that hypoxia-induced factors represent major substrates of PHD2, we generated a series of mouse strains to evaluate the relative role of HIF1α and HIF2α in regulating Treg phenotype and function. The combined loss of PHD2 and HIF2α, but not HIF1α, corrected some, but not all, abnormalities found in the PHD2$^{\Delta Treg}$ mouse strain. To uncover the mechanism whereby the PHD2-HIF2α axis regulates the capacity of Tregs to exert a homeostatic control over conventional T cells, a large transcriptomic analysis was undertaken. To be able to isolate genes specifically involved in the control of Treg activity in naive animals, we took advantage of the graded proinflammatory status of the mouse strain generated (based on colon length and spontaneous activation of Tconv cells, see *Figure 6*) to identify the gene clusters whose expression correlated with Treg-mediated immune homeostasis. This analysis led us to identify the important pathways providing mechanistic insights into the role of the PHD2-HIF2α axis in Treg biology. In particular, loss of PHD2 led to an altered expression of genes coding for chemokine receptors and adhesion molecules, suggesting a potential role of this oxygen sensor in chemotaxis and traffic. This conclusion is of particular interest in light of two observations described in this study. As previously discussed, PHD2$^{\Delta Treg}$ mice displayed a selective expansion of Th1-prone effectors in all lymphoid organs examined (*Figure 1*). Accordingly, ubiquitous loss of IFN-γ expression strongly attenuated the pro-inflammatory phenotype of mice with PHD2-deficient Tregs (*Figure 5—figure supplement 2*), thus suggesting a specific role for PHD2 in endowing Tregs to control Th1-like immune responses in vivo. Secondly, the IPA conducted on the transcriptomic data led to the identification of STAT1 as a potential common regulator of many genes whose expression was under the control of the PHD2-HIF2α axis, including in particular CXCR3 (*Hall et al., 2012*; *Koch et al., 2009*). Based on the well-described role for Treg-expressed CXCR3 in modulating Th1-like responses in vivo (*Levine et al., 2017*; *Littringer et al., 2018*) and the reduced expression of this chemokine receptor described in this study (*Figure 8e*), it is tempting to speculate that the reduced capacity of PHD2-deficient Tregs to control Th1 responses is a consequence of an altered STAT1 signaling pathway, leading to reduced CXCR3 expression. It is noteworthy that response to CXCR3 ligands has been recently shown to determine the precise positioning of effector and memory CD8 cells in peripheral lymph nodes (*Duckworth et al., 2021*). Further studies would be required to identify the precise mechanism at work since the expression of many potential chemokine receptors (including CXCR4, known to exert inhibitory function over other chemokine receptors; *Biasci et al.,*

*2020*) and adhesion molecules (such as Ly6a or CD44) appears under the control of the PHD2-HIF2α axis in Tregs. Whether altered positioning of Tregs within lymphoid organs represents an important factor contributing to the proinflammatory phenotype of PHD2$^{ΔTreg}$ mice remains, however, to be thoroughly examined. Similarly, the potential mechanistic link between HIF2α and STAT1 activation remains to be firmly established by further investigations.

Collectively, the observations reported in this study demonstrate a nonredundant role for PHD2 in controlling survival, phenotype, and the capacity of Tregs to control Th1-like responses. These biological responses appear under the control of HIF2α and are largely independent of HIF1α-regulated metabolic pathways. Although the role of the PHD2-HIF2α axis has been previously highlighted by Yamamoto and colleagues using an alternative shRNA-based approach (*Yamamoto et al., 2019*), our observations do not fully concur with the previous study on two grounds. First, in contrast to PHD2 knockdown (PHD2-KD) cells, PHD2-genetically deficient Tregs retained full suppressive capacities in vitro. Secondly, no sign of reversal to an effector state was found in PHD2-KO regulatory T cells, whereas downregulation of PHD2 expression led to an increased expression of T-bet, GATA-3, and TNFα. Notably, PHD2-KD Tregs were able to induce skin-graft rejection in the absence of bona fide effector cells, suggesting a possible acquisition of effector function by these cells. Although these observations are compatible with a possible gene-dosage effect of PHD2 on Treg biology, further studies are needed to identify the mechanism at work in these two experimental models.

In any events, both studies concur in identifying a possible deleterious role of HIF2α overactivation in the control of regulatory T cell function. These findings appear at odds with a recent publication by Tzu-Sheng Hsu and colleagues in which deletion of HIF2α, but not HIF1α, expression was found to negatively affect Treg function (*Hsu et al., 2020*). Of note, concomitant deletion of both HIF1α and HIF2α restored the suppressive activity of Tregs (*Hsu et al., 2020*). An elegant hypothesis, put forward by these authors, may help reconcile some of these apparently contradictory observations. Most experimental evidence concurs with a dual role of HIF1α in Treg differentiation and stability. In the setting of suboptimal Treg-inducing conditions, HIF1α may promote adequate expression of Foxp3 by differentiating Tregs. Once the Treg phenotype has been fully acquired, HIF1α protein expression is reduced following interaction with Foxp3 (*Clambey et al., 2012*), thus explaining the relative lack of influence of HIF1α on differentiated Tregs. As a consequence, HIF1α-KO Tregs retain full suppressive activity (*Hsu et al., 2020*). The interaction between HIF1α and Foxp3 can, however, also lead to Foxp3 protein degradation, and thus Treg instability. Therefore, forced stabilization of HIF1α (such as observed in triple PHD KOs [*Clever et al., 2016*] or pVHL-deficient Tregs [*Lee et al., 2015a*]) leads to loss of Foxp3 expression and Treg identity and acquisition of proinflammatory functions. Inflammation observed in these mouse strains can be largely attributed to the proinflammatory influence of ex-Tregs. As discussed for HIF1α, HIF2α also appears to regulate Treg stability, albeit in a different direction. Despite a normal phenotype at rest, mice displaying HIF2α-deficient Tregs were largely defective in suppressing inflammation in the gut and lungs (*Hsu et al., 2020*). This proinflammatory phenotype was largely explained by the HIF1α-dependent reprogramming of HIF2α-deficient Tregs into IL-17-secreting cells. Collectively, the available literature points to a central role for HIF1α in determining Treg stability and function in vivo. Depending on the biological pathway leading to its increased expression and/or protein stabilization, HIF1α promotes the differentiation of Tregs into IFN-γ (in triple PHD KOs or pVHL-deficient Tregs) or IL-17 (in HIF2α-deficient Tregs) secreting cells. Although the mechanism underlying the acquisition of Th1 vs. Th17-like profiles in these models remains to be established, the induction of a glycolytic metabolism is probably instrumental in mediating Treg instability (*Shi and Chi, 2019*).

In the present study, loss of HIF1α expression did not revert the phenotype of PHD2-HIF2α-deficient Tregs, despite re-establishing a control-like expression of pro-glycolytic genes (*Figure 7e*). Accordingly, PHD2-deficient Tregs did not acquire the capacity to produce proinflammatory cytokines (*Figure 1—figure supplement 1*) nor displayed any significant loss of Foxp3 expression upon in vitro culture (*Figure 2k and l*) or in vivo transfer (*Figure 4e*). Thus, the available evidence suggests that in PHD2-sufficient cells HIF2α allows adequate Treg function by negating the influence of HIF1α on Foxp3-expression, while overactivation of HIF2α activity secondary to the loss of PHD2 expression leads to altered Treg phenotype, most probably via a STAT1-dependent pathway.

Considering the specific role of PHD2, it is worth mentioning that both the transcriptomic data and our own unpublished observations (indicating an increased sensitivity of triple PHD2-HIF1α-HIF2α

Tregs-specific KO mice to chemical induced colitis) suggest that while the capacity of Tregs to control tissue homeostasis in the naive state is under the predominant control of the PHD2-HIF2α axis, other non-HIF PHD2-substrates (*Meneses and Wielockx, 2016*; *Mikhaylova et al., 2008*; *Chan et al., 2009*; *Romero-Ruiz et al., 2012*; *Huo et al., 2012*; *Xie et al., 2015*; *Lee et al., 2015b*; *Guo et al., 2016*) probably play an important role in Treg biology under strong inflammatory settings. Finally, this study suggests that some caution should be exerted in the administration of PHD inhibitors presently considered for the treatment of renal anemia (*Gupta and Wish, 2017*), inflammatory bowel diseases (*Marks et al., 2015*), as well as Parkinson's disease (*Li et al., 2018*) as these compounds may display some proinflammatory effects via the alteration of Treg phenotype and function in vivo.

# Materials and methods

## Key resources table

| Reagent type (species) or resource | Designation | Source or reference | Identifiers | Additional information |
|---|---|---|---|---|
| Genetic reagent (*Mus musculus*) | C57BL/6 | Envigo | RRID:MGI:5658455 | Horst, The Netherlands |
| Genetic reagent (*M. musculus*) | *Egln1*f/f | The Jackson Laboratory | RRID:IMSR_NM-CKO-2100497 | P. Carmeliet (VIB-KULeuven) |
| Genetic reagent (*M. musculus*) | *Foxp3*-Cre-YFP | PMID:18387831 | RRID:IMSR_JAX:016959 | A. Liston (KULeuven) |
| Genetic reagent (*M. musculus*) | *Hif1a*f/f (Hif1atm3Rsjo/J) | The Jackson Laboratory | RRID:IMSR_JAX:007561 | F. Bureau (Liege University) |
| Genetic reagent (*M. musculus*) | *Epas*f/f (Epas1tm1Mcs/J) | The Jackson Laboratory | RRID:IMSR_JAX:008407 | J.A. Lopez (Madrid University) |
| Genetic reagent (*M. musculus*) | *Ifng*-/- | The Jackson Laboratory | RRID:IMSR_CARD:178 | Bar Harbor, ME |
| Genetic reagent (*M. musculus*) | CD45.1 (B6.SJL-Ptprca Pepcb/BoyJ) | The Jackson Laboratory | RRID:IMSR_JAX:002014 | Bar Harbor, ME |
| Genetic reagent (*M. musculus*) | *Rag2*-/- | The Jackson Laboratory | RRID:IMSR_JAX:008449 | Bar Harbor, ME |
| Antibody | Anti-mouse CD278 (Icos)-biotin (C398.4A, mouse monoclonal) | eBioscience | 13-9949-82 | (1:100) |
| Antibody | Anti-mouse CD27-PeCy7 (LG.7F9, mouse monoclonal) | eBioscience | 25-0271-82 | (1:250) |
| Antibody | Anti-mouse Foxp3-FITC (FJK-16s, mouse monoclonal) | eBioscience | 71-5775-40 | (1:100) |
| Antibody | Anti-mouse RORγt-PE (B2D, mouse monoclonal) | eBioscience | 12-6981-82 | (1:100) |
| Antibody | Anti-mouse T-bet-PE (4B10, mouse monoclonal) | eBioscience | 12-5825-82 | (1:100) |
| Antibody | Anti-mouse PD1-PECF594 (J43, mouse monoclonal) | BD Biosciences | 562523; RRID:AB_2737634 | (1:100) |
| Antibody | Anti-mouse CXCR3-APC (CXCR3-173, mouse monoclonal) | BD Biosciences | 562266; RRID:AB_11153500 | (3:500) |
| Antibody | Anti-mouse CD24-PECF594 (M1/69, mouse monoclonal) | BD Biosciences | 562477; RRID:AB_11151917 | (1:100) |
| Antibody | Anti-mouse CD25-BB515 (PC61, mouse monoclonal) | BD Biosciences | 564424; RRID:AB_2738803 | (1:100) |
| Antibody | Anti-mouse CD44-PECy7 (IM7, mouse monoclonal) | BD Biosciences | 560569; RRID:AB_1727484 | (1:100) |

*Continued on next page*

*Continued*

| Reagent type (species) or resource | Designation | Source or reference | Identifiers | Additional information |
|---|---|---|---|---|
| Antibody | Anti-mouse CD4-A700 (RM4-5, mouse monoclonal) | BD Biosciences | 557956; RRID:AB_396956 | (3:500) |
| Antibody | Anti-mouse CD8-A700 (53-6.7, mouse monoclonal) | BD Biosciences | 557959; RRID:AB_396959 | (3:500) |
| Antibody | Anti-mouse CD4-PB (RM4-5, mouse monoclonal) | BD Biosciences | 558107; RRID:AB_397030 | (1:100) |
| Antibody | Anti-mouse CD62L-A700 (MEL-14, mouse monoclonal) | BD Biosciences | 560517; RRID:AB_1645210 | (1:100) |
| Antibody | Anti-mouse GATA3-PE (L50-823, mouse monoclonal) | BD Biosciences | 560074; RRID:AB_1645330 | (1:10) |
| Antibody | Anti-mouse RORγt-PECF594 (Q31-378, mouse monoclonal) | BD Biosciences | 562684; RRID:AB_2651150 | (1:200) |
| Antibody | Anti-mouse STAT1 (pY701)-A488(4a, mouse monoclonal) | BD Biosciences | 612596; RRID:AB_399879 | (1:10) |
| Antibody | Anti-mouse IFNγ-PE (XMG1.2, mouse monoclonal) | BD Biosciences | 554412; RRID:AB_395376 | (1:100) |
| Antibody | Anti-mouse IL-10-APC (JES5-16E3, mouse monoclonal) | BD Biosciences | 554468; RRID:AB_398558 | (1:100) |
| Antibody | Anti-mouse IL-17-PerCP-Cy5.5 (N49-653, mouse monoclonal) | BD Biosciences | 560799; RRID:AB_2033981 | (1:100) |
| Antibody | Anti-CD3 antibody (2c11, mouse monoclonal) | BioXCell | 145-2c11 | 20 µg/mouse |
| peptide, recombinant protein | Streptavidin-PECy7. | BD Biosciences | 557598; RRID:AB_10049577 | (1:100) |
| peptide, recombinant protein | IFN-γ protein | PeproTech | 315-05 | 50 ng/ml |
| Chemical compound, drug | Evans blue | Sigma | 314-13-6 | 0.5% |
| Chemical compound, drug | Brefeldin-A | eBioscience | 00-4506-51 | (1:1000) |
| Chemical compound, drug | Dextran sodium sulfate, colitis grade (36,000–50,000 Da) | MP Biomedical | 160110 | 2% |
| Commercial assay or kit | LIVE/DEAD kit | Invitrogen | L10119 | (1:1000) |
| Commercial assay or kit | Anti-CD90.2 beads MACS | Miltenyi | 130-121-278 | (1:5) |
| Commercial assay or kit | Anti-CD4 beads MACS | Miltenyi | 130-117-043 | (1:3) |
| Sequence-based reagent | *Egln1 (PHD2)_F* | This paper | PCR primers | AGGCTATGTCCGTCACGTTG |
| Sequence-based reagent | *Egln1 (PHD2)_R* | This paper | PCR primers | TACCTCCACTTACCTTGGCG |
| Sequence-based reagent | *Egln2 (PHD1)_F* | This paper | PCR primers | TCACGTGGACGCAGTAATCC |
| Sequence-based reagent | *Egln2 (PHD1)_R* | This paper | PCR primers | CGCCATGCACCTTAACATCC |
| Sequence-based reagent | *Egln3 (PHD3)_F* | This paper | PCR primers | AGGCAATGGTGGCTTGCTAT |
| Sequence-based reagent | *Egln3 (PHD3)_R* | This paper | PCR primers | GACCCCTCCGTGTAACTTGG |
| Sequence-based reagent | *Hif1a_F* | This paper | PCR primers | CATCAGTTGCCACTTCCCCA |
| Sequence-based reagent | *Hif1a_R* | This paper | PCR primers | GGCATCCAGAAGTTTTCTCACAC |
| Sequence-based reagent | *Epas1 (HIF2a)_F* | This paper | PCR primers | ACGGAGGTCTTCTATGAGTTGGC |
| Sequence-based reagent | *Epas1 (HIF2a)_R* | This paper | PCR primers | GTTATCCATTTGCTGGTCGGC |

*Continued on next page*

*Continued*

| Reagent type (species) or resource | Designation | Source or reference | Identifiers | Additional information |
|---|---|---|---|---|
| Sequence-based reagent | Ifng_F | This paper | PCR primers | TGCCAAGTTTGAGGTCAACA |
| Sequence-based reagent | Ifng_R | This paper | PCR primers | GAATCAGCAGCGACTCCTTT |
| Sequence-based reagent | Il12a_F | This paper | PCR primers | CCTCAGTTTGGCCAGGGTC |
| Sequence-based reagent | Il12a_R | This paper | PCR primers | CAGGTTTCGGGACTGGCTAAG |
| Sequence-based reagent | Il10_F | This paper | PCR primers | CCTGGGTGAGAAGCTGAAGA |
| Sequence-based reagent | Il10_R | This paper | PCR primers | GCTCCACTGCCTTGCTCTTA |
| Sequence-based reagent | Il17a_F | This paper | PCR primers | ATCCCTCAAAGCTCAGCGTGTC |
| Sequence-based reagent | Il17a_R | This paper | PCR primers | GGGTCTTCATTGCGGTGGAGAG |
| Sequence-based reagent | Il1b_F | This paper | PCR primers | CAAGCTTCCTTGTGCAAGTG |
| Sequence-based reagent | Il1b_R | This paper | PCR primers | AGGTGGCATTTCACAGTTGA |
| Sequence-based reagent | Il4_F | This paper | PCR primers | ATGCACGGAGATGGATGTG |
| Sequence-based reagent | Il4_R | This paper | PCR primers | AATATGCGAAGCACCTTGGA |
| Sequence-based reagent | Il6_F | This paper | PCR primers | GTTCTCTGGGAAATCGTGGA |
| Sequence-based reagent | Il6_R | This paper | PCR primers | GCAAGTGCATCATCGTTGTT |
| Sequence-based reagent | Rpl32_F | This paper | PCR primers | ACATCGGTTATGGGAGCAAC |
| Sequence-based reagent | Rpl32_R | This paper | PCR primers | TCCAGCTCCTTGACATTGT |
| Sequence-based reagent | Tnfa_F | This paper | PCR primers | GCCTCCCTCTCATCAGTTCTA |
| Sequence-based reagent | Tnfa_R | This paper | PCR primers | GCTACGACGTGGGCTACAG |
| Sequence-based reagent | Il12b_F | This paper | PCR primers | ATGTGTCCTCAGAAGCTAACC |
| Sequence-based reagent | Il12b_R | This paper | PCR primers | CTAGGATCGGACCCTGCAGGGAAC |
| Software, algorithm | Prism 6 | GraphPad | RRID:SCR_002798 | Version 6.0 |

## Mice

C57BL/6 mice were purchased from Envigo (Horst, The Netherlands). *Egln1*<sup>f/f</sup> mice were provided by P. Carmeliet (VIB-KULeuven, Leuven, Belgium); *Foxp3*-Cre-YFP mice, developed by A. Rudensky (*Rubtsov et al., 2008*), were kindly provided by A. Liston (KULeuven, Leuven, Belgium); Hif1atm3Rs-jo/J (*Hif1a*<sup>f/f</sup>) mice were kindly provided by F. Bureau (Liege University, Liege, Belgium); Epas1tm-1Mcs/J (*Epas*<sup>f/f</sup>) mice were provided by J.A. Lopez (Madrid University, Madrid, Spain); *Ifng*<sup>-/-</sup>, CD45.1 (B6.SJL-Ptprc<sup>a</sup> Pepc<sup>b</sup>/Boy<sup>J</sup>) and *Rag2*<sup>-/-</sup> mice were obtained from The Jackson Laboratory (Bar Harbor, ME). All mice were backcrossed for more than 10 generations into a C57BL/6 background and housed in individually ventilated cages. *Foxp3*-Cre-YFP mice were crossed with *Egln1*<sup>f/f</sup>, *Hif1a*<sup>f/f</sup>, *Epas*<sup>f/f</sup> to produce mice with Treg-specific deletion of PHD2, HIF1α, HIF2α, PHD2-HIF1α, PHD2-HIF2α, or PHD2-HIF1α-HIF2α. Heterozygous *Foxp3*<sup>cre/+</sup> *Egnl1*<sup>fl/fl</sup> mice were generated by crossing the *Foxp3*-Cre-YFP mice with *Egln1*<sup>f/f</sup> mice. All mice were used between 8 and 14 weeks of age. PHD2-sufficient mice (expressing *Foxp3*-Cre-YFP, or floxed forms of PHD2, HIF1α, and HIF2α-encoding alleles and generated as littermates in our colony) were used as appropriate controls in the early stages of this work. These mice displayed a phenotype indistinguishable from WT mice and were therefore considered as a single experimental group throughout this study in order to reach statistical significance in all experiments. The experiments were carried out in compliance with the relevant laws and institutional guidelines and were approved by the Université Libre de Bruxelles Institutional Animal Care and Use Committee (protocol number CEBEA-4).

## Antibodies, intracellular staining, and flow cytometry

The following monoclonal antibodies were purchased from eBioscience: CD278 (ICOS)-biotin, CD27-PeCy7, Foxp3-FITC, RORγt-PE, T-bet-PE; or from BD Biosciences: PD1-PECF594, CXCR3-APC,

CD24-PECF594, CD25-BB515, CD44-PECy7, CD4-A700, CD8-A700, CD4-PB, CD62L-A700, GATA3-PE, RORγt-PECF594, STAT1 (pY701)-A488, IFNγ-PE, IL-10-APC, IL-17-PerCP-Cy5.5, streptavidin-PECy7. Live/dead fixable near-IR stain (Thermo Fisher) was used to exclude dead cells. For transcription factor staining, cells were stained for surface markers, followed by fixation and permeabilization before nuclear factor staining according to the manufacturer's protocol (Foxp3 staining buffer set from eBioscience). For cytokine staining, cells were stimulated in media containing phorbol 12-myristate 13-acetate (50 ng/ml, Sigma-Aldrich), ionomycin (250 ng/ml, Sigma-Aldrich), and brefeldin-A (1/100, eBioscience) for 3 hr. After stimulation, cells were stained for surface markers, followed by fixation and permeabilization before intracellular staining according to the manufacturer's protocol (cytokine staining buffer set from BD Biosciences). For phosphorylation staining, cells were stimulated with IFN-γ (50 ng/ml, PeproTech) for 30 min, fixed with formaldehyde, and permeabilized with methanol before staining. Flow cytometric analysis was performed on a Canto II (BD Biosciences) and analyzed using FlowJo software (Tree Star).

## T cell cultures

After removal of Peyer's patches and mesenteric fat, intestinal tissues were washed in Hank's balanced salt solution (HBSS) 3% fetal calf serum (FCS) and phosphate-buffered saline (PBS), cut in small sections, and incubated in HBSS 3% FCS containing 2.5 mM EDTA and 72.5 µg/ml DTT for 30 min at 37°C with agitation to remove epithelial cells, and then minced and dissociated in RPMI containing liberase (20 µg/ml, Roche) and DNase (400 µg/ml, Roche) at 37°C for 30 min. Leukocytes were collected after a 30% Percoll gradient (GE Healthcare). Lymph nodes and spleens were mechanically disrupted in culture medium. CD4+ T cells were positively selected from organ cell suspensions by magnetic-activated cell sorting using CD4 beads (MACS, Miltenyi) according to the manufacturer's protocol and purified as CD4+ CD44loCD62LhiCD25− or CD4+ CD44lo CD62Lhi YFP− by fluorescence-activated cell sorting. T cells were cultured at 37°C in RPMI supplemented with 5% heat-inactivated fetal bovine serum (Sigma-Aldrich), 1% nonessential amino acids (Invitrogen), 1 mM sodium pyruvate (Invitrogen), 2 mM L-glutamine (Invitrogen), 500 U/ml penicillin/500 µg/ml streptomycin (Invitrogen), and 50 µM β-mercaptoethanol (Sigma-Aldrich). To generate iTreg cells, cells were cultured in 24-well plates coated with 5 µg/ml anti-CD3 (BioXCell, clone 145-2C11) at 37°C for 72 hr. The culture was supplemented with anti-CD28 (1 µg/ml, BioXCell, clone 37.51), TGF-β (3 ng/ml, eBioscience), and IL-2 (10 ng/ml, PeproTech) for optimal iTreg cell polarization.

## Treg cell suppression assays

### In vitro assay

CD4+ CD44loCD62Lhi CD25− naive T cells were isolated from the spleen of CD45.1+ mice by cell sorting after positive enrichment for CD4+ cells using MACS LS columns (Miltenyi) and labeled with carboxyfluorescein diacetate succinimidyl ester (CFSE, Thermo Fisher). CD4+YFP+ Treg cells were isolated from the spleen of Foxp3cre or PHD2ΔTreg mice by cell sorting. Splenocytes from wild-type B6 mice were depleted in T cells (anti-CD90.2 beads, MACS, Miltenyi) using MACS LS columns (Miltenyi) and used as feeder cells. $4 \times 10^4$ CFSE-labeled naive T cells were cultured for 72 hr with feeder cells ($1 \times 10^5$) and soluble anti-CD3 (0,5 µg/ml) in the presence or absence of various numbers of Treg cells as indicated.

### In vivo assay

Rag2-/- mice were injected i.v. with a mixture of naive, CFSE-labeled, CD4+ T cells (CD45.1+ CD4+ CD44lo CD62Lhi CD25−) ($1 \times 10^6$) and splenic Treg from Foxp3cre or PHD2ΔTreg mice ($3.3 \times 10^5$). Six days after the injection, *Rag2*-/- mice were sacrificed and CD4+ T cells proliferation and activation analyzed by flow cytometry.

## DSS-induced colitis

Foxp3cre or PHD2ΔTreg mice were provided with 2% DSS (MP Biomedical, 160110) in tap water for 5 days. On day 5, the DSS-containing water was replaced with normal drinking water and mice were followed during 14 days for body weight, survival, and colitis severity. Colitis severity score was assessed by examining weight loss, feces consistency, and hematochezia (Hemoccult SENSA, Mckesson Medical-Surgical, 625078) as described in *Kim et al., 2012*. Colon samples were washed with PBS and rolled

from the distal to proximal end, transected with a needle and secured by bending the end of the needle and fixed in fresh 4% paraformaldehyde (Sigma-Aldrich) overnight and further subjected to routine histological procedures for embedment in paraffin and hematoxylin and eosin (H&E) staining. Tissues were analyzed and scored in a blinded fashion by an independent histopathologist, and representative images were subsequently chosen to illustrate key histological findings.

## Toxoplasma infection

ME-49 type II *T. gondii* was kindly provided by Dr De Craeye (Scientific Institute of Public Health, Belgium) and was used for the production of tissue cysts in C57BL/6 mice previously (1–3 months) inoculated with three cysts by gavage. Animals were killed, and the brains were removed. Tissue cysts were counted, and mice were infected by intragastric gavage with 10 cysts. Mice were sacrificed at day 8 after infection.

## Anti-CD3 mAb-induced enteritis

Mice were injected i.p. with a CD3-specific antibody (clone 145-2C11, BioXCell 20 µg/mouse) on days 0 and 2 and weighted daily. Mice were sacrificed on day 3 and cytokine production evaluated by qPCR as indicated in the figure legend.

## Hematological analysis

Mice blood was obtained from the submandibular vein and collected into heparin-prefilled tubes. Blood samples were analyzed using a Sysmex KX-21 N Automated Hematology Analyzer.

## Evans blue assay

Blood vessel permeability was assessed as previously described (*Radu and Chernoff, 2013*). Briefly, 200 µl of a 0.5% sterile solution of Evans blue (Sigma) in PBS was i.v. injected in mice. After 30 min, organs were collected, weighted, and were put in formamide. After 24 hr in a 55°C water bath, absorbance was measured at 600 nm.

## RT-qPCR

RNA was extracted using the TRIzol method (Invitrogen) and reverse transcribed with Superscript II reverse transcriptase (Invitrogen) according to the manufacturer's instructions. Quantitative real-time RT-PCR was performed using the SYBR Green Master mix kit (Thermo Fisher). Primer sequences were as follows:

*Rpl32* (F) ACATCGGTTATGGGAGCAAC; *Rpl32* (R) TCCAGCTCCTTGACATTGT; *Il1b* (F) CAAGCTTCCTTGTGCAAGTG; *Il1b* (R) AGGTGGCATTTCACAGTTGA; *Il10* (F) CCTGGGT-GAGAAGCTGAAGA; *Il10* (R) GCTCCACTGCCTTGCTCTTA; *Ifng* (F) TGCCAAGTTTGAGGTCAACA; *Ifng* (R) GAATCAGCAGCGACTCCTTT; *Il6* (F) GTTCTCTGGGAAATCGTGGA; *Il6* (R) GCAAGTG-CATCATCGTTGTT; *Il17a* (F) ATCCCTCAAAGCTCAGCGTGTC; *Il17a* (R) GGGTCTTCATTGCGGT GGAGAG; *Il12a* (F) CCTCAGTTTGGCCAGGGTC; *Il12a* (R) CAGGTTTCGGGACTGGCTAAG; *Il12b* (F) ATGTGTCCTCAGAAGCTAACC; *Il12b* (R) CTAGGATCGGACCCTGCAGGGAAC; *Tnfa* (F) GCCTCCCT CTCATCAGTTCTA; *Tnfa* (R) GCTACGACGTGGGCTACAG; *Egln1* (F) AGGCTATGTCCGTCACGTTG; *Egln1* (R) TACCTCCACTTACCTTGGCG; *Hif1a* (F) CATCAGTTGCCACTTCCCCA; *Hif1a* (R) GGCA TCCAGAAGTTTTCTCACAC; *Epas1* (F) ACGGAGGTCTTCTATGAGTTGGC; *Epas1* (R) GTTATCCA TTTGCTGGTCGGC.

## RNA-sequencing and analysis

All RNA-seq analyses were performed using ≥2 biological replicates. Total RNA was prepared from purified splenic Treg cells using the TRIzol method (Invitrogen). 200 ng of total RNA was subsequently used to prepare RNA-seq library by using TruSeq RNA sample prep kit (Illumina) according to the manufacturer's instructions. Paired-end RNA-sequencing was performed on a NovaSeq 6000 (Illumina) (BRIGHTcore joint facility, ULB-VUB, Brussels, Belgium). Sequenced reads were aligned to the mouse genome (NCBI37/mm9), and uniquely mapped reads were used to calculate gene expression. Data analysis was performed using R program (DESeq2 package). Differentially expressed genes are considered significant when the false discovery rate (FDR or adjusted p-value) < 0.05 and the $\log_2$ fold change (FC) > 0.5. Upstream regulators analysis was performed following IPA. IPA predicts functional

regulatory networks from gene expression data and provides a significance score (p-value) for each network according to the fit of the network to the set of genes in the database.

## Statistical analysis

All statistical analyses were conducted using GraphPad Prism (GraphPad Software). Statistical difference between two groups was determined by an unpaired, two-tailed Student's *t*-tests. A one-way or two-way ANOVA was used for multigroup comparisons together with Tukey's multiple comparisons post hoc tests. Survival significance in DSS-induced colitis was determined by a log-rank Mantel–Cox test. Data is judged to be statistically significant when p-value<0.05. In figures, asterisks denote statistical significance (*p<0.05; **p<0.01; ***p<0.001; ****p<0.0001).

## Acknowledgements

We thank Valérie Acolty, Caroline Abdelaziz, and Véronique Dissy for animal care and technical support. The development of mouse models of toxoplasmosis would not have been possible without the kind assistance of Guillaume Oldenhove. This work was supported by the European Regional Development Fund (ERDF) and the Walloon Region (Wallonia-Biomed portfolio, 411132-957270), a grant from the Fonds Jean Brachet and research credit from the National Fund for Scientific Research, FNRS, Belgium. FA is a Research Associate at the FNRS. YA is recipient of a research fellowship from the FNRS/Télévie. HH has been supported by a Belgian FRIA fellowship.

## Additional information

### Funding

| Funder | Grant reference number | Author |
|---|---|---|
| European Regional Development Fund | | Yousra Ajouaou<br>Hind Hussein<br>Fabienne Andris<br>Muriel Moser<br>Stanislas Goriely<br>Oberdan Leo |
| Fond de la recherche scientifique | | Yousra Ajouaou<br>Hind Hussein |
| Walloon region | | Fabienne Andris<br>Muriel Moser<br>Stanislas Goriely<br>Oberdan Leo |
| Fond Jean Brachet | | Fabienne Andris<br>Muriel Moser<br>Stanislas Goriely<br>Oberdan Leo |

The funders had no role in study design, data collection and interpretation, or the decision to submit the work for publication.

### Author contributions

Yousra Ajouaou, Conceptualization, Data curation, Formal analysis, Funding acquisition, Investigation, Methodology, Validation, Visualization, Writing - original draft, Writing - review and editing; Abdulkader Azouz, Data curation, Formal analysis, Software, Visualization, Writing - review and editing; Anaëlle Taquin, Investigation, Methodology, She performed additional experiments and contributed to data analysis, She performed additional experiments and contributed to data analysis; Sebastien Denanglaire, Mohammad Krayem, Methodology; Hind Hussein, Methodology, She performed additional experiments and contributed to data analysis, She performed additional experiments and contributed to data analysis; Fabienne Andris, Supervision, Writing - review and editing; Muriel Moser, Funding acquisition, Project administration, Resources, Supervision, Writing - review and editing; Stanislas Goriely, Project administration, Resources, Writing - review and editing; Oberdan

Leo, Conceptualization, Funding acquisition, Methodology, Project administration, Resources, Supervision, Validation, Writing - original draft, Writing - review and editing

## Author ORCIDs
Yousra Ajouaou (iD) http://orcid.org/0000-0002-1734-6505
Fabienne Andris (iD) http://orcid.org/0000-0002-8644-1020
Oberdan Leo (iD) http://orcid.org/0000-0002-3621-4743

## Ethics

The experiments were performed in compliance with the relevant laws and institutional guidelines and were approved by the Local Ethic Committee. We received specific approval for this study from the Université Libre de Bruxelles Institutional Animal Care and Use Committee (protocol numbers CEBEA-4 and 31).

## Decision letter and Author response
Decision letter https://doi.org/10.7554/eLife.70555.sa1
Author response https://doi.org/10.7554/eLife.70555.sa2

## Additional files

### Supplementary files
• Transparent reporting form

### Data availability
Sequencing data have been deposited in GEO under accession code GSE184581. Numerical data used to generate the figures have been provided as source data files.

The following dataset was generated:

| Author(s) | Year | Dataset title | Dataset URL | Database and Identifier |
| --- | --- | --- | --- | --- |
| Ajouaou Y, Azouz A, Taquin A, Hussein H, Andris F, Moser M, Goriely S, Leo O | 2021 | The oxygen sensor Prolyl hydroxylase domain 2 regulates the in vivo suppressive capacity of regulatory T cells | https://www.ncbi.nlm.nih.gov/geo/query/acc.cgi?acc=GSE184581 | NCBI Gene Expression Omnibus, GSE184581 |

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
