## [Editor Report]

The possibility that hypoxia signaling pathways exert control over immunity by actions on regulatory T lymphocytes is of great interest given that immune inflammatory pathology is often associated with hypoxia. This article supports the cell-intrinsic role of a specific isoform of hypoxia-inducible factor 2 (HIF2) in reducing the function of these cells when HIF is activated by inactivation of its regulatory hydroxylase. Aside from the biology, this is important because both HIF activators and specific HIF-2 inhibitors are in use clinically.

---

## [Decision Letter]

**Decision letter after peer review:**

Thank you for submitting your article "The oxygen sensor Prolyl hydroxylase domain 2 regulates the in vivo suppressive capacity of regulatory T cells" for consideration by *eLife*. Your article has been reviewed by 3 peer reviewers, and the evaluation has been overseen by a Reviewing Editor and Mone Zaidi as the Senior Editor. The following individuals involved in review of your submission have agreed to reveal their identity: Chris Pugh (Reviewer #1); Randall Johnson (Reviewer #2).

Essential revisions:

Your manuscript has now been carefully reviewed for *eLife* and the reviewers have debated their somewhat different perspectives.

You will see from the correspondence that though all considered that the manuscript contains useful data on the important question as to whether activity of the HIF hydroxylase system directs regulatory T cell function in accordance with oxygen availability. Nevertheless, there were differences of opinion on the extent to which your findings advance knowledge on this question.

My summary of this is that the overall view (and my own view) is that there is merit in publishing the current extensive datasets that you have obtained, not least in the interests of consolidating findings in a complex but important area. We ask however that you address all specific questions raised by the reviewers as part of the publication process.

Decision to publish would be subject to satisfactory responses. In particular, it is essential that you address the following with changes to the manuscript:

(i) For proper presentation to a general audience, it is necessary that you present the existing published literature that is directly relevant to your work in the introduction. This enables your results to be understood in the context of that literature. Your discussion may then concentrate on differences/similarities/extensions of that work by your own data and on whether the overarching question of the role of physiological/pathological signalling in directing T-reg activity has or has not been answered by the literature in total.

(ii) An important part of the novelty is dependent on restriction of recombination using the Foxp3-Cre-YFP line. More data is needed to show that recombination is indeed restricted as proposed, or that additional sites of recombination might contribute to the phenotype if they exist.

(iii) It is necessary to provide more detailed description of the haemorragic abdomen – in particular, whether this might imply a more complex phenotype affecting other components of haematopoiesis.

(iv) All reviewers felt more data was required to support the mechanistic hypotheses. However, it was acknowledged that this would likely demand an unrealistic amount of further work. The authors should therefore either provide any further experiment support they can, or moderate the mechanistic claims. In particular, they should address the issue raised by one of the reviewers that changes in splenic architecture could have confounded interpretation of the importance of lymphocyte mislocalization in that organ.

*Reviewer #1 (Recommendations for the authors):*

Please add more description of the existing literature describing effects of disruption of HIF-signalling on Treg function to the Introduction and identify (and discuss possible explanations for) similarities and differences between these reports and your own data in the Discussion.

Please clarify expression of YFP outside the Treg lineage and any changes in Phd2 expression in non-lymphoid tissues (including endothelium, liver and colon). Please comment on the relative changes in red and white pulp of the spleen and provide data to show whether haematocrit has or has not changed in the Phd2-deficient Treg mice.

Please specify the sex of animals used in all the experiments.

Please indicate absolute cell counts as well as percentages in the various lymphoid organs.

It would be nice if critical gating strategies were shown to support the flow cytometry analyses.

I think there are a number of places where the authors could indicate some caveats to their interpretation of their results, many of which should be apparent from my public review.

I would like to consider further specific recommendations to Authors in the light of discussion of the public review with my co-reviewers and the Reviewing Editor.

Typos and issues of clarity

P4 para 1 'prolin-hydroxylation' should be 'prolyl-hydroxylation'

In describing Figure S1 the statement is made 'Spontaneous expression of GLUT1, a well-known target of HIF1α, was also only found in Foxp3-expressing cells in these mice, further supporting the selective depletion of PHD2 in Tregs (Figure S1).' Is the effect not one of up-regulation rather than expression?

The authors' surnames should be completed in reference 22.

Availability of raw data.

Please make the raw data for the work presented in Figure 6 and all the Supplementary Figures publically available. Please make it clear how to access all aspects of publically available data in the text.

*Reviewer #2 (Recommendations for the authors):*

Despite this not being necessary for the publication of this manuscript, my only suggestion to the authors is to test effect of chemical PHD inhibitor drugs on TREG function and use the specific HIF2 inhibitor PT-2385 to try and abolish any potential side effect. Additionally, the use of PT2385 per se in any of the in vivo assays with the TREG PHD2 KO cells would not only rule out any cellular adaptation mechanism that might occur during TREG development in double or triple KOs and would be relevant for the clinic as this inhibitor is being tested in humans.

*Reviewer #3 (Recommendations for the authors):*

1. The authors should analyze the early Treg development in the thymus.

2. Extrinsic vs. intrinsic effect should be addressed using Foxp3-cre +/- mice, or co-adoptive transfer approaches.

3. The authors should perform more experiments to clarify the mis-localization vs. direct suppression in vivo. The current model does not explain the effector/TH1-like phenotype.

4. More mechanistic experiments are required to address the PHD2-HIF2s relation to the Stat1 phosphorylation, and the mis-localization issue.

---

## [Author Response]

Decision to publish would be subject to satisfactory responses. In particular, it is essential that you address the following with changes to the manuscript:(i) For proper presentation to a general audience, it is necessary that you present the existing published literature that is directly relevant to your work in the introduction. This enables your results to be understood in the context of that literature. Your discussion may then concentrate on differences/similarities/extensions of that work by your own data and on whether the overarching question of the role of physiological/pathological signalling in directing T-reg activity has or has not been answered by the literature in total.

We have followed this recommendation and provided additional information on the “state of the art” and previous work covering the topic of our work. This is now included in the Introduction section.

(ii) An important part of the novelty is dependent on restriction of recombination using the Foxp3-Cre-YFP line. More data is needed to show that recombination is indeed restricted as proposed, or that additional sites of recombination might contribute to the phenotype if they exist.

This point is well taken. As previously stated in our previous version, expression of the Cre-allele appears as highly restricted to the Foxp3^+^ subset of lymphoid cells. As expected however, we have found and reported in Figure 8-figure supplement 1 that a minor, but identifiable subset of non-lymphoid (CD45-negative) cells (from 1 to 5%, depending on the organ considered) expresses the Cre allele. Despite our efforts, we could not identify the nature of these cells, and their contribution to the observed phenotype remains therefore to be established. We do however believe that (i) the fact that PHD2-KO Tregs display a reduced suppressive capacity when co-transferred with naïve T cells in Rag2-deficient mice (Figure 4) and (ii) the observations performed in heterozygous Foxp3^Cre/+^ Egln1^fl/fl^ (Figure 3) demonstrating that loss of PHD2 expression affects the Treg phenotype in a cell-autonomous, intrinsic fashion, strongly indicate that PHD2-expression is required for adequate Treg phenotype and function irrespectively of the environment in which they differentiate.

(iii) It is necessary to provide more detailed description of the haemorragic abdomen – in particular, whether this might imply a more complex phenotype affecting other components of haematopoiesis.

We have been able to demonstrate that these mice display an increased haematocrit and enhanced vascular permeability that could be a consequence of enhanced Vegfa expression by Tregs. At present, the mechanism leading to these haematological consequences remain to be established. Notably however, an haemorrhagic abdomen was not observed in all mice examined (albeit in a majority of them, approximatively 70%), whereas the phenotypical alterations of Tregs and the enhanced activation of Tconvs were confirmed in all animals displaying PHD2-deficient Tregs, once again suggesting that the effect of PHD2 on Treg biology are not a consequence of altered haematological parameters.

(iv) All reviewers felt more data was required to support the mechanistic hypotheses. However, it was acknowledged that this would likely demand an unrealistic amount of further work. The authors should therefore either provide any further experiment support they can, or moderate the mechanistic claims. In particular, they should address the issue raised by one of the reviewers that changes in splenic architecture could have confounded interpretation of the importance of lymphocyte mislocalization in that organ.

This point is well taken, and we have now deleted the last figure of the manuscript and provided a less speculative conclusion to our study. We pointed to a possible link between the PHD2-HIF2α axis and STAT1 activation as an important factor regulating Treg phenotype and function.

Reviewer #1 (Recommendations for the authors):Please add more description of the existing literature describing effects of disruption of HIF-signalling on Treg function to the Introduction and identify (and discuss possible explanations for) similarities and differences between these reports and your own data in the Discussion.

We believe we have adequately responded to this request

Please clarify expression of YFP outside the Treg lineage and any changes in Phd2 expression in non-lymphoid tissues (including endothelium, liver and colon). Please comment on the relative changes in red and white pulp of the spleen and provide data to show whether haematocrit has or has not changed in the Phd2-deficient Treg mice.

We have performed the requested experiments but could not achieve a definite conclusion regarding the cause of the haematological abnormalities observed in PHD2^ΔTregs^ mice.

Please indicate absolute cell counts as well as percentages in the various lymphoid organs.

An example of absolute cells counts is now provided in the Figure 1—figure supplement 4.

It would be nice if critical gating strategies were shown to support the flow cytometry analyses.

An example of the gating strategy used throughout this study is now provided in Figure 1—figure supplement 3

Availability of raw data.

All raw data are now available, see Excel Source data.

Reviewer #2 (Recommendations for the authors):Despite this not being necessary for the publication of this manuscript, my only suggestion to the authors is to test effect of chemical PHD inhibitor drugs on TREG function and use the specific HIF2 inhibitor PT-2385 to try and abolish any potential side effect. Additionally, the use of PT2385 per se in any of the in vivo assays with the TREG PHD2 KO cells would not only rule out any cellular adaptation mechanism that might occur during TREG development in double or triple KOs and would be relevant for the clinic as this inhibitor is being tested in humans.

We thank the reviewer for this interesting suggestion that we will take into consideration for our future work

Reviewer #3 (Recommendations for the authors):1. The authors should analyze the early Treg development in the thymus.

We thank you the reviewer for this insightful suggestion. We have now analysed the frequency of thymic Treg precursors (see Figure 2 b-d) and found an increased frequency of immature Tregs displaying the Foxp3^lo^ CD25^neg^ phenotype in the thymus of PHD2^ΔTregs^ mice. Consistent with the expected tissue distribution of the Cre allele, no differences were found in the frequency of the Foxp3^neg^ CD25^pos^ subset of precursor Tregs. Overall, this analysis suggests a role for PHD2 in thymic Treg development and a possible “immature-like” state of peripheral Tregs in PHD2^ΔTregs^ mice.

2. Extrinsic vs. intrinsic effect should be addressed using Foxp3-cre +/- mice, or co-adoptive transfer approaches.

We thank again the reviewer for this suggestion. The results from these experiments are now depicted in Figure 3, and demonstrate that the phenotypic alterations found in PHD2-deficient Tregs appear as “cell autonomous”. Moreover, this experiment also strongly suggest that PHD2-deficient T cells display reduced fitness when compared to their WT counterpart, a finding that we have been able to confirm in adoptive co-transfer experiments (data not shown).

3. The authors should perform more experiments to clarify the mis-localization vs. direct suppression in vivo. The current model does not explain the effector/TH1-like phenotype.

This comment is well taken, and as previously discussed (see remark n°8), we are now convinced that our work does not provide sufficient arguments to sustain the “mis-localisation” hypothesis, partly due to the unresolved question of the profound changes in spleen cell morphology observed in PHD2^ΔTregs^ mice. We have therefore omitted this set of data from the present version of the manuscript.

4. More mechanistic experiments are required to address the PHD2-HIF2s relation to the Stat1 phosphorylation, and the mis-localization issue.

We fully agree with this final remark, but the amount of work that would be required to experimentally connect HIF2α to STAT1 activation is probably beyond reach within the time allowed for this review. In particular it appears, from the literature, that STAT1 regulation is a complex phenomenon involving a set of post-translational modifications (including methylation and palmitoylation) that could represent targets of HIF2α-mediated regulation.